# Environmental eustress modulates β-ARs/CCL2 axis to induce anti-tumor immunity and sensitize immunotherapy against liver cancer in mice

Chaobao Liu [1,3,4], Yang Yang[1,4], Cheng Chen [1,4], Ling Li[1,4], Jingquan Li[1], Xiaonan Wang[1], Qiao Chu[1], Lin Qiu[2], Qian Ba [1], Xiaoguang Li [1✉] & Hui Wang[1,2✉]

Although psycho-social stress is a well-known factor that contributes to the development of cancer, it remains largely unclear whether and how environmental eustress influences malignant diseases and regulates cancer-related therapeutic responses. Using an established eustress model, we demonstrate that mice living in an enriched environment (EE) are protected from carcinogen-induced liver neoplasia and transplantable syngeneic liver tumors, owning to a CD8$^+$ T cell-dependent tumor control. We identify a peripheral Neuro-Endocrine-Immune pathway in eustress, including Sympathetic nervous system (SNS)/β-adrenergic receptors (β-ARs)/CCL2 that relieves tumor immunosuppression and overcomes PD-L1 resistance to immunotherapy. Notably, EE activates peripheral SNS and β-ARs signaling in tumor cells and tumor infiltrated myeloid cells, leading to suppression of CCL2 expression and activation of anti-tumor immunity. Either blockade of CCL2/CCR2 or β-AR signaling in EE mice lose the tumor protection capability. Our study reveals that environmental eustress via EE stimulates anti-tumor immunity, resulting in more efficient tumor control and a better outcome of immunotherapy.

[1] State Key Laboratory of Oncogenes and Related Genes, Center for Single-Cell Omics, School of Public Health, Shanghai Jiao Tong University School of Medicine, Shanghai 200025, China. [2] CAS Key Laboratory of Nutrition, Metabolism and Food safety, Shanghai Institute of Nutrition and Health, University of Chinese Academy of Sciences, Chinese Academy of Sciences, Shanghai, China. [3] School of Basic Medical Sciences, Fudan University, Shanghai 200032, China. [4] These authors contributed equally: Chaobao Liu, Yang Yang, Cheng Chen, Ling Li. ✉email: lixg@shsmu.edu.cn; huiwang@shsmu.edu.cn

The impact of psychological stress on health has been widely concerned and investigated[1–3], especially after the COVID-19 outbreak that led to increased fear, anxiety, depression, and feelings of stress among the community[4,5]. Epidemiologic studies have shown a connection between psychological stress and cancer progression[6–10]. However, limited and inconsistent evidence was provided. Moreover, the majority of these studies were focused on stressors associated with negative aspects of severe stress (distress), while the impact of more positive environment stimulations (eustress) on cancer biology is still largely unknown.

Psychological stress causes immune dysfunction which subsequently regulates the initiation and progression of tumors[11,12]. Many studies have shown that immune cells, including lymphocytes and myeloid cells, harbor receptors essential to stress hormone response[11,13]. The sympathetic nervous system (SNS) regulates response to stress, including eustress and distress. Activation of SNS has advantageous and harmful effects on organisms depending on the duration of response and other unknown factors. The systemic feature of sympathetic response suggests that tumor, immune, or stromal cells in the tumor microenvironment might be affected by changes in stress-associated hormone levels from a distance, or even locally[13].

The liver is one of the organs in charge of dealing with physiological stress. Population studies have revealed a positive causal relationship of psychological distress on three major hepatic diseases: viral hepatitis, cirrhosis, and hepatocellular carcinoma (HCC)[14,15]. However, whether the positive mental attributes reduce the risk of liver cancer, whether they are beneficial to the outcome of cancer-related therapies, and the underlying molecular mechanisms remain unclear. From a clinical point of view, a better understanding of how stress (distress and eustress) alters hepatic diseases would provide additional tools for treating patients with liver diseases. It would improve the quality of life in patients by optimizing hospitalization conditions and ensure a more efficient therapeutic approach.

Environment Enrichment (EE) is an established environmental eustress model for providing mice more social interaction in a large activity space with running wheels for sports and other toys for hiding and playing[16–18]. It has been demonstrated that EE results in a series of beneficial effects that make the mice "happier", reduces anxiety[19], as well as improves the symptoms of neurodegenerative diseases[20] and cancers[16,17]. However, it requires further investigation and interrogation on whether and how eustress influences liver cancer and cancer-related therapeutic responses.

In this work, using three carcinogen-induced liver neoplasia models and three transplantable syngeneic liver tumors models, we showed that EE-mediated environmental eustress-induced tumors are primarily dependent on the adaptive immunity via modulating local β-ARs/CCL2 axis in the tumor microenvironment.

## Results

**Housing in enriched environment reduces hepatic carcinogenesis and tumor growth.** Psychosocial distress is a risk factor for the incidence and death of liver diseases[15]. Reduced distress levels and anxiety-like behaviors are the two main phenotypes provided by the established model of eustress, such as enriched environment (EE) housing (Fig. 1A). To investigate the potential protective role of eustress on liver malignant progression, we took advantage of three carcinogen- or diet-induced liver neoplasia mouse models, including diethylnitrosamine (DEN) and carbon tetrachloride (CCl$_4$) model, DEN and high-fat diet (HFD) model, CCl$_4$ model and three transplantable murine liver tumors mouse

models (Hepa1-6, LPC-H12, H22) that were housed in a standard environment (SE) or EE. The DEN + CCl$_4$ mouse model was induced by a single injection of DEN followed by repeated administration of CCl$_4$, mimicking genotoxic injury and advanced fibrosis-associated HCC in humans (Fig. 1B). In SE mice, a dramatic potentiation of the liver tumor was observed following administration of DEN/CCl$_4$, with 100% of mice developed liver tumors at 5 months of age. In comparison, 72% of EE mice developed liver tumor with reduced tumor numbers and size (Fig. 1C–E). Histopathology analysis further indicated that EE facilitated the protection of mice against the DEN/CCl$_4$-induced liver fibrosis, hepatocellular adenoma, and carcinoma (Fig. 1F, G). It is noteworthy that EE housing also reduced serum AST and ALT (Supplementary Fig. 1A, B), pathological markers associated with the consequence of chronic liver injury. We also explored the impact of EE on DEN/high-fat diet (HFD)-induced (Supplementary Fig. 1C) or CCl$_4$-induced (Supplementary Fig. 1G) liver carcinogenesis, mimicking the steatohepatitis-HCC model and the chronic inflammation-HCC model, respectively. EE consistently decreased the tumor numbers and tumor size in the DEN/HFD model (Supplementary Fig. 1D–F) and in the CCl$_4$-induced HCC model (Supplementary Fig. 1H–K), in addition to an increase of the body weight (Supplementary Fig. 1L).

Next, we used transplantable syngeneic Hepa1-6 and H22 liver tumor models to investigate the influence of EE on tumor growth and progression. Three-week-old C57BL/6 mice (for Hepa1-6 model) and BALB/c mice (for H22 model) were randomized either to stay in SE or EE housing condition for 3 weeks, and then received subcutaneous injections of Hepa1-6 and H22 cells, respectively (Fig. 1H, K). Mice were kept housing in their original homes (Fig. 1H, K). In EE mice, Hepa1-6 and H22 tumor growth was much slower than that in SE housing, with inhibitions of 49.3 and 50.7%, respectively (Fig. 1I, L). The tumor mass was reduced by 68.4 and 47.4%, respectively (Fig. 1I–M).

Taken together, these results demonstrated a protective impact of EE against HCC initiation and growth.

**EE mediated-antitumor effect depends on CD8$^+$ T cells.** Positive eustress via EE has been suspected of boosting the host immune system against cancer[16,21,22]. Here, we asked whether the immune system is associated with the EE-mediated antitumor effect in HCC models. To investigate whether and what type of immune effector cells mediate EE–induced tumor inhibition, tumor-infiltrated immune cells were analyzed in the DEN + CCl$_4$-induced tumor (Fig. 2A), Hepa1-6 tumor (Fig. 2B), and H22 tumor (Fig. 2C). In EE mice, flow cytometric analysis of DEN + CCl$_4$ tumors showed a significant increase of CD8$^+$ T cells and a dramatic decrease of M-MDSCs, G-MDSCs, and M2-like TAMs compared to those in SE mice (Fig. 2A, D). The numbers of other immune cells, such as CD4$^+$ T cells, regulatory T cells (Treg), and natural killer cells (NK) exhibited no appreciable changes (Fig. 2A and Supplementary Fig. 2G, H). Similarly, in EE mice bearing transplantable Hepa1-6 or H22 tumor, an increase of CD8$^+$ T cell and reduced G-MDSCs and M2-like TAMs were detected (Fig. 2B, C, E). Immunohistochemistry staining of CD4, CD8, F4/80and Ly6G on DEN + CCl$_4$ tumors further confirmed the reshaping of the tumor microenvironment (Supplementary Fig. 2A, B). To test whether the adaptive immune system is essential for EE-mediated tumor protection, mice were treated with either anti-CD4, anti-CD8, or anti-NK1.1 depletion antibodies in conjunction with SE or EE housing. While EE was still responsible for tumor growth after CD4$^+$ T cell and NK cell depletion (Supplementary Fig. 2C, F), CD8$^+$ T cells depletion abolished the antitumor effect of EE on Hepa1-6 (Fig. 2F), H22 (Fig. 2G) LPC-H12 (Fig. 2H), and DEN + CCl$_4$ (Fig. 2I) tumor

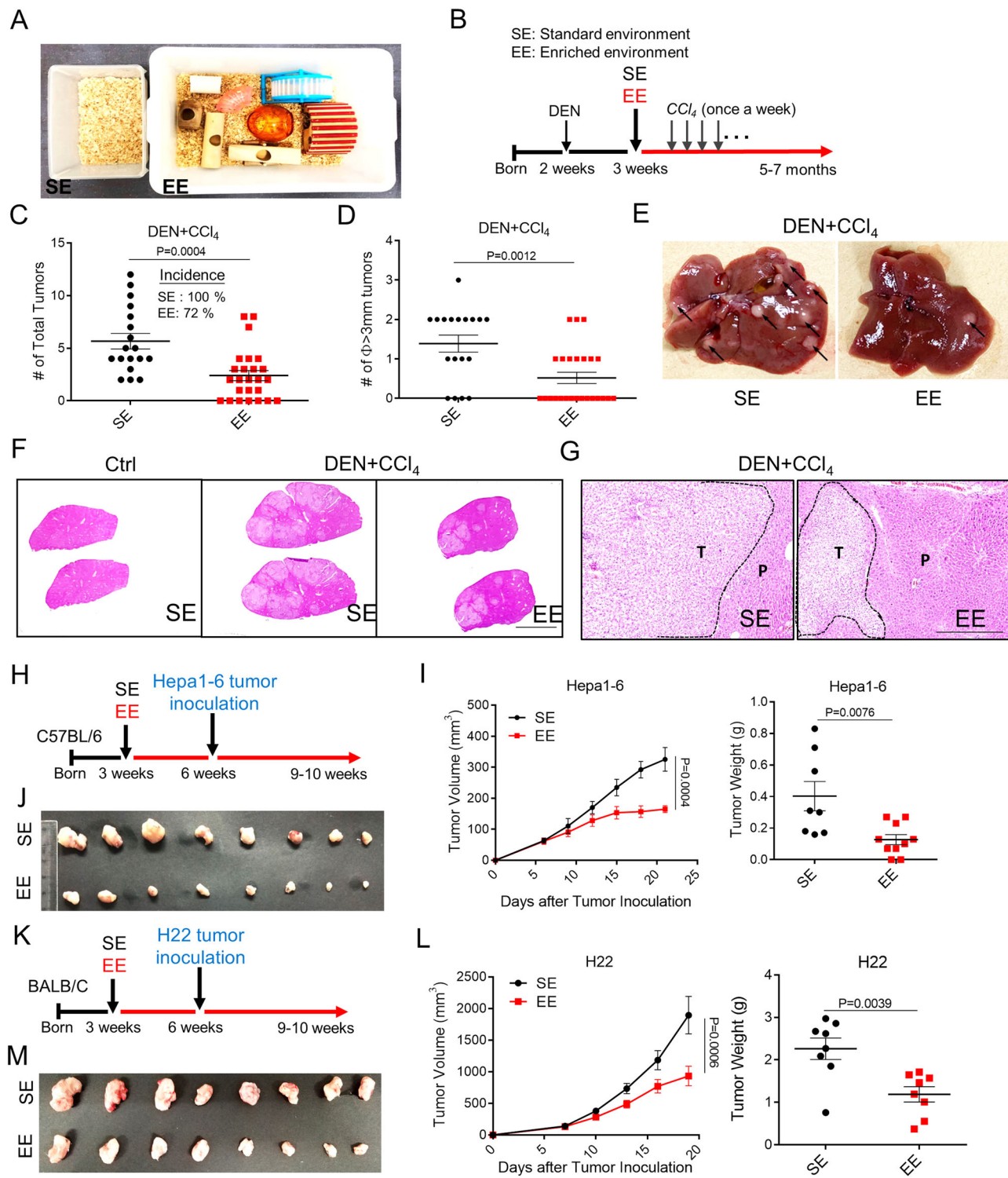

models. This data suggests that CD8$^+$ T cells, rather than CD4$^+$ T or NK cells, are required for EE-mediated tumor repression.

TAMs and MDSCs, the two major subpopulations of immunosuppressive myeloid-derived cells, are known to impair the infiltration and proliferation of functional T cells in the tumor microenvironment. The results above raised the question of whether the increase of CD8$^+$ T cells is due to EE-mediated suppression of TAMs and G-MDSCs infiltration. As expected, in SE mice, either depletion of TAMs with clodronate liposomes (Supplementary Fig. 2D) or G-MDSCs with Ly6G-antibody (Supplementary Fig. 2E) inhibited the tumor growth and induced

the infiltration of CD8$^+$ T cell (Supplementary Fig. 3D), mimicking the action of EE (Supplementary Fig. 2D, E). Mice with TAMs or G-MDSCs depletion had a higher percentage of CD8$^+$ T cells in tumors, and there was no difference between the SE and EE (Supplementary Fig. 3D). These results suggested that EE increased CD8$^+$ T cells by suppressing TAMs and G-MDSCs infiltration.

In EE mice, qRT-PCR analysis of Hepa1-6 tumors showed a significant increase of proinflammatory cytokine expression (IL-12a, iNOS, IL-6, TNF, CD14, IFN-gamma, IFN-beta) and a dramatic decrease of anti-inflammatory cytokine expression

**Fig. 1 Environmental eustress reduces HCC tumorigenesis and progression. A** The cage for Standard Environment (SE) and Enriched Enrichment (EE) feeding conditions, which were kept consistent for all experiments. **B** Schematic representation of the experimental design and timeline for DEN/CCl₄-induced tumor model. **C–E** Total tumor number **C** and diameter ø ≥ 3 mm tumor number **D** in the liver of DEN + CCl₄-treated mice with SE or EE-feeding conditions, and a representative image of the liver with tumors for each group was shown **E** Arrowheads indicated the tumor nodules. (n = 18 for SE group; n = 25 for EE group). All data are presented as the mean ± SEM, and analyzed by two-tailed unpaired Student's t test. **F, G** Typical H&E-staining-based histopathology images of livers from normal mice or DEN/CCl₄-induced tumors-bearing mice with SE or EE-feeding conditions (**F**, Scale bar, 0.85 cm). The HCC tumor region was indicated with dotted line (Scale bar, 100 μm). T tumor region, P para-tumor region. **H–J** Scheme of experimental procedure for subcutaneous Hepa1-6 murine HCC tumor model **H**. Tumor volume of tumor-bearing mice fed under SE or EE conditions (**I**, left). Tumor mass (**I**, right) and representative images **J** were shown at 23 days after tumor implantation. All data are presented as the mean ± SEM, and analyzed by two-tailed unpaired Student's t test. **K–M** Experimental design for subcutaneous H22 murine HCC tumor model **K**. Tumor volume of mice fed under SE or EE (**L**, left). Tumor mass (**L**, right) and representative image **M** were shown at 20 days after tumor implantation. All data are presented as the mean ± SEM, and analyzed by two-tailed unpaired Student's t test.

(Arg1, mMGL2, Fiz1, CD163, Retn1a, IL-10, and TGF-beta) compared to those in SE mice (Supplementary Fig. 3A). To explore the influence of EE on CD8$^+$ T cells through G-MDSCs and macrophages, primary CD8$^+$ T cells were cocultured with polarized G-MDSCs or macrophages in vitro, which were isolated from femurs of wild-type mice or subcutaneous layer from Hepa1-6 tumor-burden mice under SE or EE housing condition (Supplementary Fig. 3B, C). The results showed EE-primed G-MDSCs or macrophages isolated from tumor-burdened mice increased CD8$^+$ T cell proliferation, indicating EE abolished the immunosuppressive role of tumor-educated G-MDSCs or M2-like macrophages (Supplementary Fig. 3B, C).

The results above clarified that the EE reshaped the immunosuppressive tumor microenvironment and enhanced the CD8$^+$ T cell-mediated antitumor immunity.

**CCL2/CCR2 signaling is required for the EE-induced antitumor immunity.** A growing number of studies have identified that environmental eustress and distress play a significant role in influencing the inflammatory response[11,23]. Next, we established the models of wild-type mice or HCC-burdened mice under SE or EE housing and compared the scale and intensity of immune response in the context of 25 inflammatory cytokines and chemokines profiles (Fig. 3A and Supplementary Fig. 4A, B). In SE housing, analysis of the blood serum identified increased levels of chemokines CCL2, CCL3, CCL4, CXCL1, CXCL2, G-CSF, and cytokines IL-1β in DEN/CCl₄ treated mice compared to wild-type mice (Fig. 3A). Interestingly, EE housing has no apparent influence on the level of serum inflammatory factors in wild-type mice but significantly reduced the level of DEN/CCl₄-stimulated CCL2 (Fig. 3A). CCL4 and G-CSF were also decreased, although without statistical significance (Fig. 3A). Similarly, a dramatic reduction of CCL2 was also observed in the blood serum or tumor tissues of CCl₄-induced (Supplementary Fig. 4A, C), and DEN/HFD-induced mouse model (Supplementary Fig. 4B, D) housing in the EE condition compared to that in SE. CCL2 has been proved to be the major chemokine for tumor-infiltrated immunosuppressive myeloid cells, such as TAMs and MDSCs, contributing to liver fibrosis, steatosis, and hepatocarcinogenesis[24,25]. In line with this, reduced CCL2 expression in the tumor tissue was found in the DEN/CCl₄-induced (Fig. 3B, Supplementary Fig. 4G, 4E), CCl₄-induced (Supplementary Fig. 4C) and DEN/HFD-induced HCC model (Supplementary Fig. 4D) under EE housing condition compared to SE. We further examined whether CCL2/CCR2 signaling was required for the antitumor effect of EE. Strikingly, the protective effect of EE against DEN/CCl₄-induced hepatocarcinogenesis was abrogated in CCR2$^{-/-}$ mice (Fig. 3C). To further determine the role of CCL2/CCR2 signaling on EE-mediated antitumor immunity, we investigated tumor-infiltrated immune cells in DEN/CCl₄-induced tumor by flow cytometry analysis. In WT

mice housing in EE, CD8$^+$ T cells were remarkably increased, and M2-like TAMs and G-MDSCs were decreased in the tumor microenvironment, while in CCR2$^{-/-}$ mice, no apparent changes of CD8$^+$ T cells and M2-like TAMs were found between the SE and EE housing conditions (Fig. 3D). Of note, G-MDSCs were still decreased in the tumor tissue from CCR2$^{-/-}$ mice housing in EE (Fig. 3D). Next, we recurred to transplantable Hepa1-6 and H22 tumor models to confirm the tumor-suppressive effect of EE dependent on CCL2 modulation. High levels of CCL2 were excreted in Hepa1-6 cells, while there was barely excreted in H22 cells (Supplementary Fig. 4E). The analysis indicated that EE facilitated the reduction of CCL2 expression in Hepa1-6 tumor tissue with consistency (Supplementary Fig. 4F, G and Fig. 4E). We further used a neutralized antibody against mouse CCL2 to inhibit its function. As shown in Fig. 3F, treatment with anti-CCL2 antibody significantly inhibited the tumor growth of Hepa1-6 (Fig. 3F) in SE mice. To be noted, blockade of CCL2 abrogated the EE-induced inhibitory effect on tumor growth in DEN + CCl4 (Fig. 3E) and Hepa1-6 models (Fig. 3F), confirming the suppression role of CCL2 in the EE-induced tumor. Flow cytometry analysis further revealed that blocking CCL2 in SE mice mimicked the effect of EE-mediated reshaping of the tumor microenvironment, that is, an increase of CD8$^+$ T cells and M1-like TAMs and a decrease of G-MDSCs and M2-like TAM (Fig. 3G).

To better confirm whether tumor cell or host CCL2 signaling was essential for EE-mediated tumor inhibition, WT or CCL2$^{-/-}$ mice housing in SE or EE were inoculated with Hepa1-6 cells of different CCL2 status (CCL2 WT or CRISPR-mediated KO). The results showed that the knockout of tumor-derived CCL2 or host-derived CCL2 could suppress tumor growth and partially diminish the antitumor effect of EE. In contrast, blockade of both tumor-derived and host-derived CCL2 signaling abolished the EE's protective functions (Supplementary Fig. 4G). In line with this, in the CCL2-null H22 tumor model, EE still had a tumor protective effect, which was abrogated after blockade of the host CCL2 signaling with CCL2 neutralized antibody (Fig. 3H).

Taken together, these data indicated that CCL2 might be the key to EE-induced antitumor immunity and reshaping of the tumor microenvironment.

**EE enhances β-AR signaling to control tumor via the sympathetic nervous system.** SNS is commonly associated with stress response. To establish a mechanistic link among EE, SNS, and antitumor immunity, we first investigated the influence of EE on norepinephrine (NE) and epinephrine (EPI), which were the major neuroendocrine mediators in peripheral SNS activation. The results showed increased NE and EPI levels in serum in both wild-type mice (Fig. 4A) and DEN + CCl₄-induced tumor-bearing mice (Fig. 4B) in the EE housing condition. In normal mouse livers, the level of NE under EE housing was elevated and

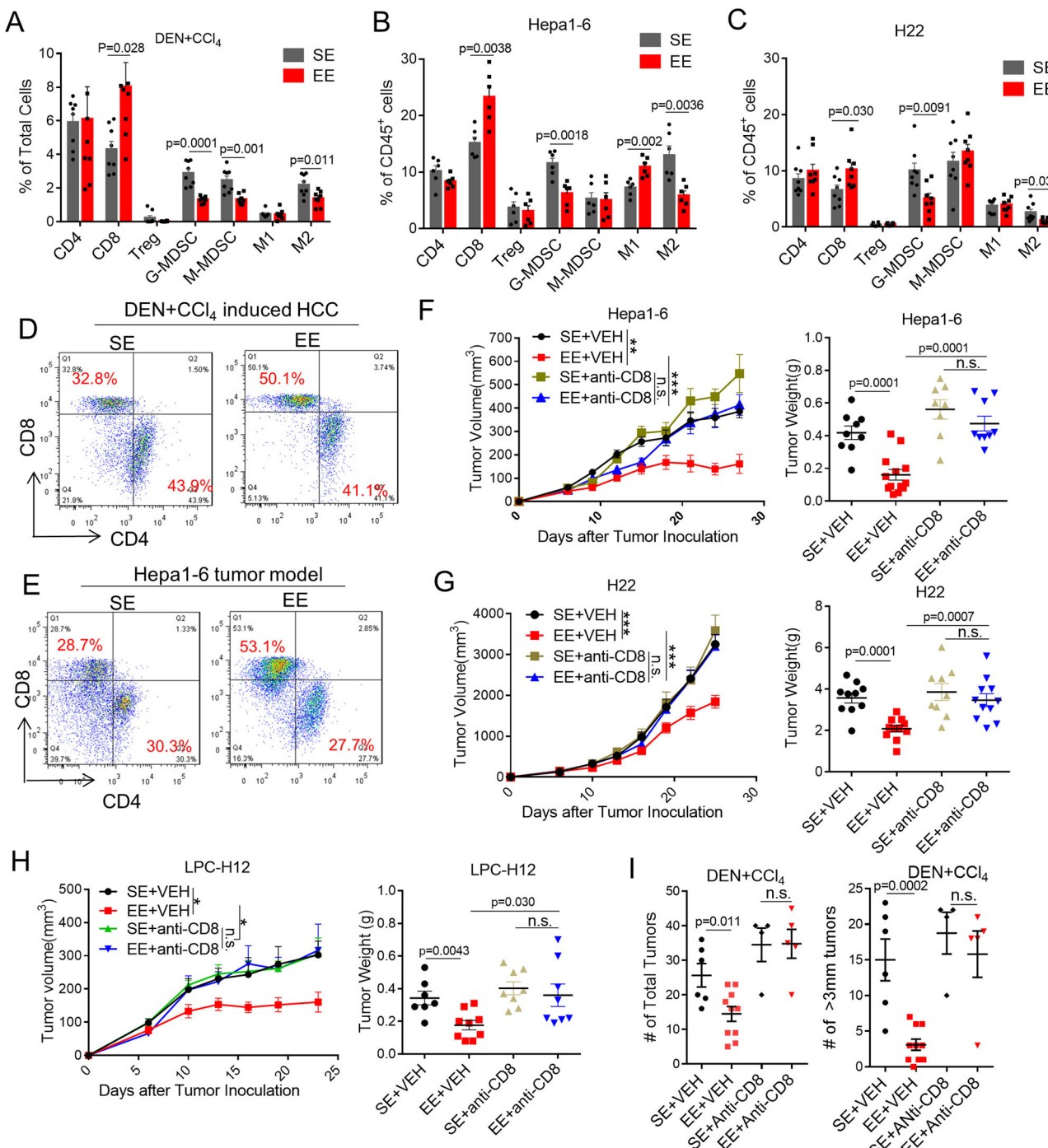

**Fig. 2 Environmental eustress reshapes tumor microenvironment and reduces tumor growth dependent on CD8+ T cells. A–C** Different infiltrated immune cells within tumor microenvironment were analyzed by flow cytometry from mice tumors in DEN + CCl$_4$ model **A**, and Hepa1-6 model **B**, and H22 model **C** ($n = 8$ for each group). SE standard environment, EE enriched environment. All data are presented as the mean ± SEM, and analyzed by two-tailed unpaired Student's $t$ test. **D, E** Representative flow cytometric analyses of CD4+ and CD8+ T cells in tumor tissues from DEN + CCl$_4$-induced tumor model **D** and Hepa1-6 tumor model **E**. **F–H** Tumor volume and tumor weight of subcutaneous Hepa1-6 **F**, H22 **G**, and LPC-H12 **H** tumors in mice treated with anti-CD8 neutralization antibody (anti-CD8) or vehicle (VEH) under SE or EE-feeding conditions (SE + VEH: $n = 9$, EE + VEH: $n = 13$, SE + anti-CD8: $n = 8$, EE + anti-CD8: $n = 9$ for Hepa1-6 groups; SE + VEH: $n = 10$, EE + VEH: $n = 11$, SE + anti-CD8: $n = 9$, EE + anti-CD8: $n = 11$ for H22 groups; SE + VEH: $n = 7$, EE + VEH: $n = 9$, SE + anti-CD8: $n = 8$, EE + anti-CD8: $n = 8$ for LPC-H12 groups). All data are presented as the mean ± SEM, and analyzed by two-tailed unpaired Student's $t$ test with n.s., $p > 0.05$; *$p < 0.05$; **$p < 0.01$, ***$p < 0.001$. **I** Total tumor number and diameter ø ≥ 3 mm tumor number in the liver of DEN + CCl$_4$-treated mice with CD8+ T cells depletion under SE or EE-feeding conditions (SE + VEH: $n = 6$, EE + VEH: $n = 10$, SE + anti-CD8: $n = 4$, EE + anti-CD8: $n = 5$). All data are presented as the mean ± SEM, and analyzed by two-tailed unpaired Student's $t$ test.

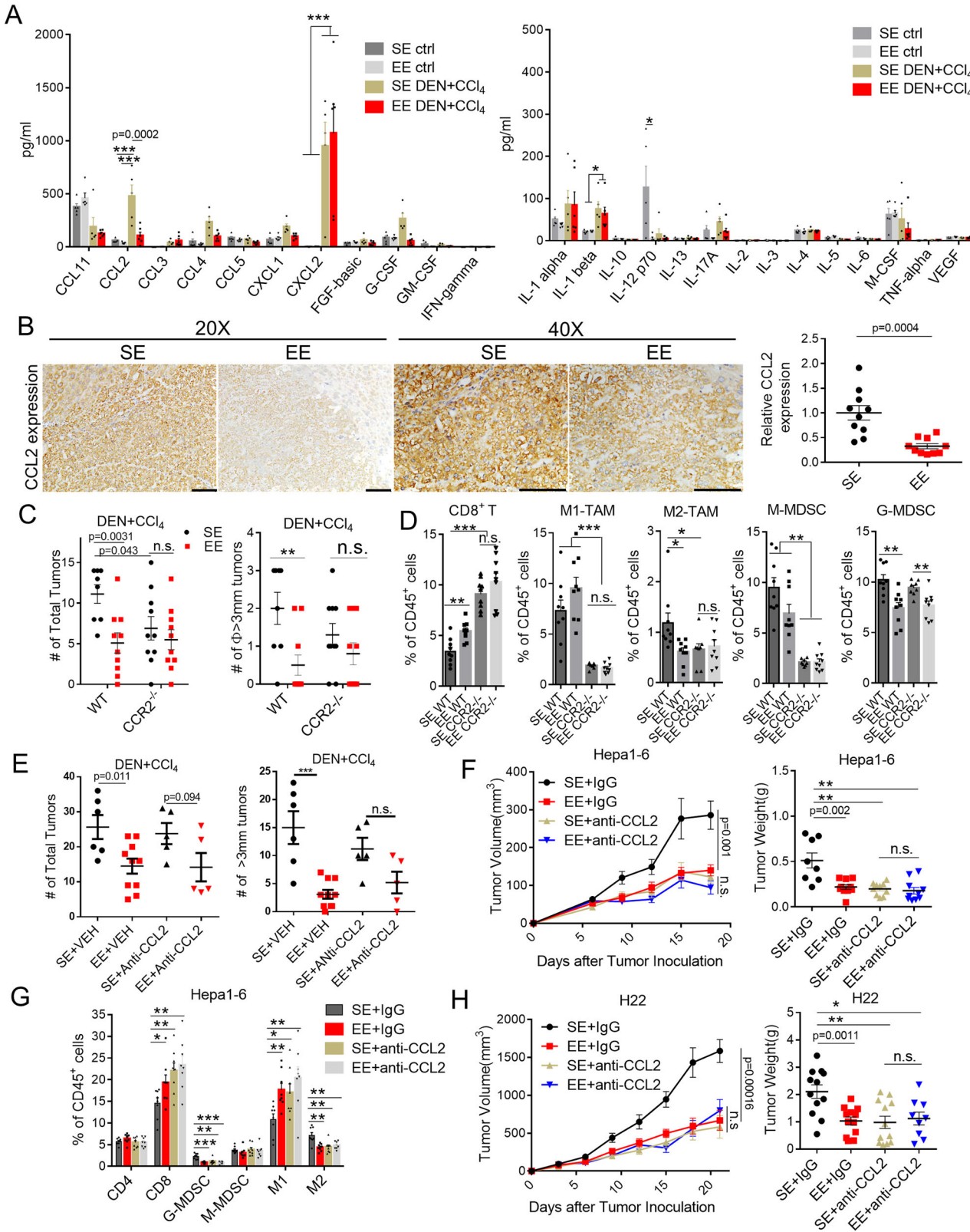

the level of EPI did not change (Fig. 4A) compared to SE housing. Whereas, in the livers of DEN + CCl4-treated mice, the levels of NE and EPI were significantly increased in the EE group (Fig. 4B). Meanwhile, EE-induced a higher NE level in the bone marrow and a higher EPI level in the spleen in DEN + CCL4-induced tumor-bearing mice (Supplementary Fig. 5B), where EE did not affect EPI or NE in the bone marrow and spleen of normal mice (Supplementary Fig. 5A). SNS is likely to directly innervate target organs relevant to cancer biology and local immune regulations. Next, we analyzed the expression of main NE and EPI receptors, adrenergic receptors (ARs) in tumor sites, including α-adrenoceptor (α1-AR or ADRA1, α2-AR, or ADRA2) and β-adrenoceptors (β1-AR or ADRB1, β2-AR or ADRB2, β3-AR or ADRB3). The data showed that mRNA expressions of

**Fig. 3 CCL2/CCR2 signaling is required for the EE-induced antitumor immunity. A** Cytokine and chemokine profiles in serum detected from Normal or DEN/CCl$_4$-induced HCC bearing mice ($n = 5$) under Standard Environment (SE) and Enriched Enrichment (EE) feeding conditions by use of Mouse Magnetic Luminex Screening Assay. All data are presented as the mean ± SEM, and analyzed by two-way ANOVA with *$p < 0.05$, **$p < 0.01$, ***$p < 0.001$. **B** Representative image of immunostaining of CCL2 on liver tumor tissues from DEN/CCl$_4$-induced HCC model (left). The relative CCL2 expression in 20 × 10 magnification was quantified by Image J analysis (right, $n = 10$). Original magnification 20 × 10, Scale bar, 100 μm; Original magnification 40 × 10, Scale bar, 100 μm.Data are presented as the mean ± SEM, and analyzed by two-tailed unpaired Student's $t$ test. **C** Total tumor number (left) and diameter ø ≥ 3 mm tumor number (right) in Wild type (WT) or CCR2 knock out (CCR2$^{-/-}$) mice injected with DEN + CCl$_4$ and fed in SE or EE conditions ($n = 8$ for WT mice in SE group, $n = 10$ for other groups). All data are presented as the mean ± SEM, and analyzed by two-tailed unpaired Student's $t$ test. **D** Infiltrated immune cells within tumor microenvironment were analyzed by flow cytometry from tumors in DEN + CCL$^4$ model ($n = 8$-9). All data are presented as the mean ± SEM, and analyzed by two-tailed unpaired Student's $t$ test with *$p < 0.05$, **$p < 0.01$, ***$p < 0.001$. **E** Total tumor number and diameter ø ≥ 3 mm tumor number in the liver of DEN + CCl$_4$-treated mice with anti-CCL2 neutralization antibody (anti-CCL2) under SE or EE-feeding conditions (SE + VEH: $n = 6$, EE + VEH: $n = 10$, SE + anti-CCL2: $n = 5$, EE + anti-CCL2: $n = 5$). All data are presented as the mean ± SEM, and analyzed by two-tailed unpaired Student's $t$ test. **F** Tumor volume (left) and tumor weight (right) of subcutaneous Hepa1-6 tumors in C57BL/6 mice treated with anti-CCL2 neutralized antibody 18 days after tumor implantation ($n = 8$ for vehicle in SE group, $n = 10$ for other groups). All data are presented as the mean ± SEM, and analyzed by two-tailed unpaired Student's $t$ test with **$p < 0.01$. **G** Infiltrated immune cells in tumor microenvironment were analyzed by flow cytometry from Hepa1-6 tumors ($n = 8$ for each group). All data are presented as the mean ± SEM, and analyzed by two-tailed unpaired Student's $t$ test with *$p < 0.05$, **$p < 0.01$, ***$p < 0.001$. **H** Tumor volume (left) and tumor weight (right) of subcutaneous H22 tumors in C57BL/6 mice treated with anti-CCL2 neutralization antibody 21 days after tumor implantation ($n = 9$ for EE + anti-CCL2 group, $n = 12$ for other groups). All data are presented as the mean ± SEM, and analyzed by two-tailed unpaired Student's $t$ test with *$p < 0.05$, **$p < 0.01$.

β1-AR and β3-AR were significantly increased in DEN + CCl$_4$-induced (Supplementary Fig. 5C) and DEN + HFD-induced tumor tissues (Supplementary Fig. 5D) in mice from EE housing. Interestingly, mRNA expression of tyrosine hydroxylase (TH), the marker of SNS activation was also upregulated in both models (Supplementary Fig. 5C, D). Histopathology immunostaining and western blotting assays further confirmed the upregulation of β1-AR, β2-AR, and β3-AR protein in the DEN + CCl$_4$-induced tumor from EE mice (Fig. 4C–E). It has been suggested that EE led to chronic activation of β-adrenergic signaling to mediate its tumor protective effect[16,17,21]. In line with this, EE failed to protect mice against DEN + CCl$_4$-induced HCC tumorigenesis when receiving the β-blocker (PROP + SR, blocking all β-ARs) (Fig. 4F). Next, we resorted to transplantable Hepa1-6, H22, and LPC-H12 subcutaneous models. Consistently, mRNA and protein expression of β1-AR, β2-AR, and β3-AR also increased (Fig. 4E and Supplementary Fig. 5E) in Hepa1-6 tumor tissues in EE housing mice. Of note, the expression of β-ARs was separately determined in the tumor cells, TAMs, and G-MDSCs in the tumor microenvironment. The results revealed that EE elevated the β1-AR and β3-AR mRNA expression levels in CD45- (mostly tumor cells), CD45+ (immune cells) (Supplementary Fig. 5F), TAMs (Supplementary Fig. 5G), and G-MDSCs (Supplementary Fig. 5H). β2-AR mRNA seemed to be only upregulated in tumor cells and TAMs but not G-MDSCs (Supplementary Fig. 5F–H). Moreover, the blockade of β-AR signaling effectively abolished the EE-mediated tumor inhibition in Hepa1-6 (Fig. 4G and Supplementary Fig. 5J) and H22 tumor models (Fig. 4H and Supplementary Fig. 5 J). In comparison, EE still had the inhibitory effect on tumor growth when α-AR signaling was blocked (Supplementary Fig. 5I). We hypothesized that sympathetic nerves are likely to transport the EE eustress signal to activate β-AR signaling to reshape the tumor microenvironment and control tumor growth. To test this hypothesis directly, we treated SE or EE housing mice with 6-hydroxydopamine (6OHDA; intraperitoneal injection) to destroy peripheral SNS, and then inoculated Hepa1-6 tumor cells. The results revealed that similarly to β-AR blocker treatment, 6OHDA-induced sympathectomy eliminated the suppressive effect of EE on tumor growth (Fig. 4I and Supplementary Fig. 5J). Consistently, flow cytometry analysis also indicated that blockade of β-ARs signaling abrogated the EE-mediated increase of CD8$^+$ T cells and decreased G-MDSCs and M2-like TAMs in the tumor microenvironment of either DEN + CCl$_4$ tumor model (Fig. 4J) or Hepa1-6 subcutaneous

tumor model (Fig. 4K). Moreover, 6OHDA-induced sympathectomy resulted in a pattern of tumor infiltrated immune cells similarly to β-ARs blocker (Fig. 4L). We then examined whether prolonged activation of β-ARs signaling via NE or EPI administration could mimic the immunomodulatory effects of EE. The results showed that moderate NE or EPI stimulation in SE mice exhibited an inhibitory effect on tumor growth (Supplementary Fig. 5K), mimicking the antitumor role of EE. Flow cytometry analysis revealed an increase of CD8$^+$ T cells and decrease of G-MDSCs and M2-like TAMs in SE mice after β-ARs signaling activation, as an "EE-like" phenotype (Supplementary Fig. 5L).

Altogether, these results demonstrated a critical role of SNS-activated β-ARs signaling in the tumor microenvironment, contributing to EE-induced tumor control.

**Activation of β-ARs signaling inhibits CCL2 expression and release**. Given that environmental eustress specifically reduced CCL2 levels and activated β-ARs signaling in the tumor microenvironment and given the essences of two signaling in EE-mediated tumor control, we decided to focus on the relevance of these signaling. Moreover, previous studies showed evidence that inflammation is regulated by sympathetic tone via β-ARs[13]. First, we evaluated whether CCL2 could be colocalized with β-ARs and analyzed their regulation mechanism using pharmacological approaches. We analyzed the expression of CCL2, β1-AR, β2-AR, β3-AR, and CD45 using in situ immunostaining in DEN + CCl$_4$-induced tumor tissue (Fig. 5A and Supplementary Fig. 6A). The results revealed the extensive-expression of β1-AR, β2-AR, and β3-AR in both the tumor and immune cells (Fig. 5A). Moreover, CCL2, colocalized with β-ARs, was also expressed in both the tumor and immune cells (Fig. 5A). It was consistent with our finding that both the tumor-derived and immune cells-derived CCL2 signaling were required for EE-induced antitumor immunity. In addition, blockade of β-ARs signaling (PROP + SR, blocking all β-ARs) dramatically abrogated the EE-induced CCL2 reduction in the blood (Fig. 5D) and tumor tissue (Fig. 5B, E, F) in DEN + CCl4-induced HCC model. Next, we resorted to the transplantable Hepa1-6 tumor models to confirm the relevance of SNS/β-ARs signaling and CCL2 regulations. Mice housing in SE or EE were subcutaneously inoculated with Hepa1-6 cells highly expressed CCL2. As shown, EE housing significantly reduced the expression level of CCL2 in the serum (Supplementary Fig. 6B) and tumor tissues (Fig. 5C), the effect of which was subverted by

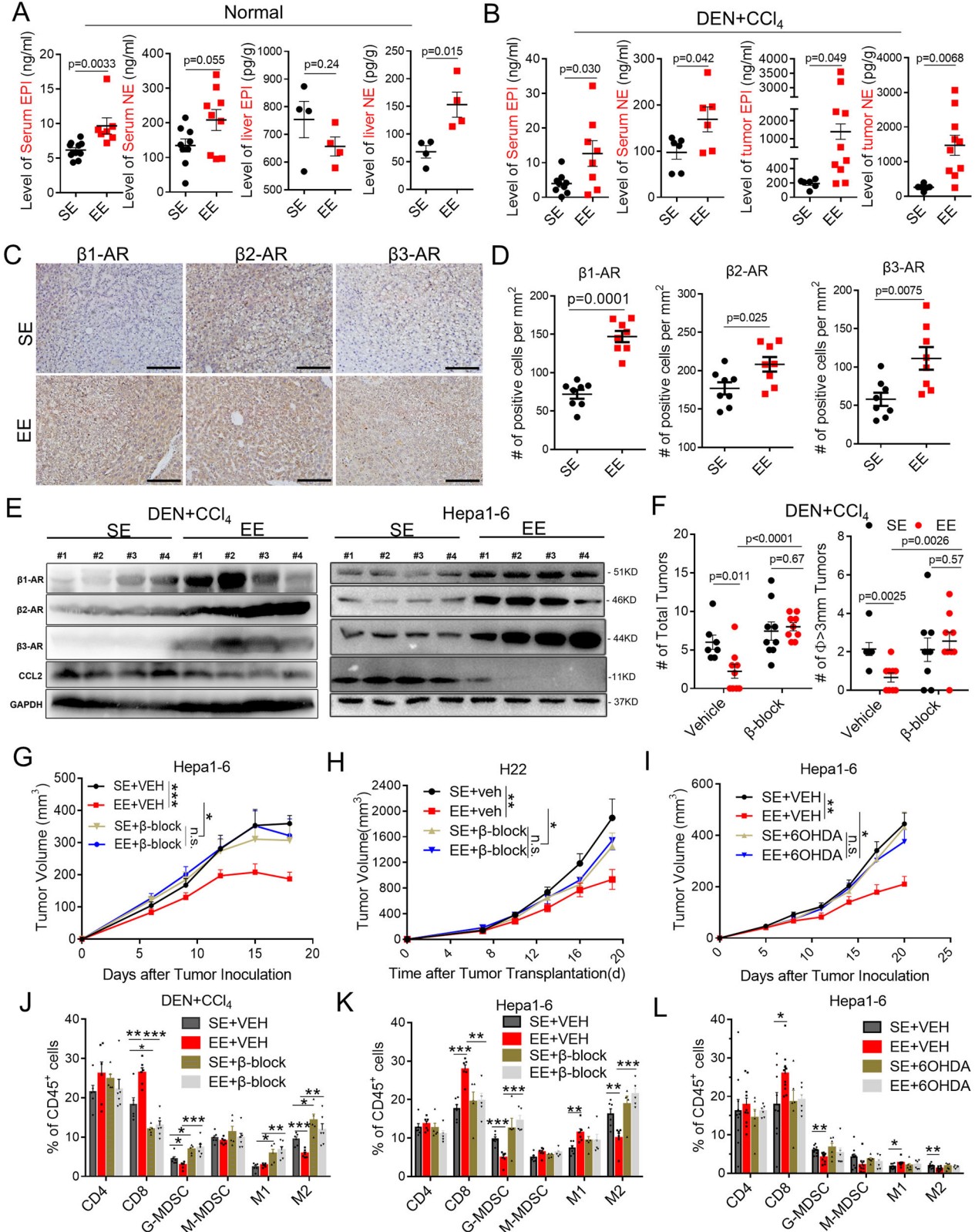

blockade of β-ARs signaling (Fig. 5C and Supplementary Fig. 6B). Interestingly, in the CCL2-null H22 tumor model, pathological immunostaining analysis showed a robust infiltration of CCL2 highly expressed macrophage-like cells in the tumor tissues from SE mice. In contrast, we observed a notable decrease of these cells in EE mice but not in EE mice with β-ARs blockade (Supplementary Fig. 6C). Next, we tested the effect of peripheral SNS on CCL2 expression in

tumor sites using 6OHDA-induced sympathectomy. Similar to β-AR blocker, 6OHDA treatment also abolished the EE-induced CCL2 suppression in the Hepa1-6 tumor tissue (Supplementary Fig. 6D). To further determine the specific expression of tumor-derived and host-derived CCL2, we isolated CD45[−] cells, CD45[+] immune cells, CD11b[+] F4/80[+]Ly6G[−] TAMs, and CD11b[+] F480[-]Ly6G[+] Ly6C[-/low] G-MDSCs from the tumor tissue of Hepa1-6

**Fig. 4 EE enhances β–AR signaling to control tumor via sympathetic nervous system. A, B** ELISA analysis of Epinephrine (EPI) and Norepinephrine (NE) in the serum and liver from normal mice **A** or DEN + CCl₄-treated mice **B** which were fed under Standard Environment (SE) and Enriched Enrichment (EE) conditions (A, level of serum EPI, SE: $n = 11$, EE: $n = 7$; level of serum NE, $n = 9$ for each group; level of liver EPI or NE, $n = 4$ for each group; B, level of serum EPI, SE: $n = 9$, EE: $n = 8$; level of serum NE, $n = 6$ for each group; level of tumor EPI or NE, SE: $n = 6$, EE: $n = 10$). All data are presented as the mean ± SEM, and analyzed by two-tailed unpaired Student's t test. **C** Representative images of immunostaining of β1-Adrenergic Receptor (AR), β2-AR, and β3-AR on liver tumors tissues from DEN + CCl₄-induced HCC mouse model. Original magnification 20 × 10, Scale bar, 100 μm. **D** Positive β1-AR, β2-AR, and β3-AR areas of liver tumor tissues were quantified by ImageJ analysis in DEN + CCl₄-induced HCC mouse model ($n = 8$ for each group). All data are presented as the mean ± SEM, and analyzed by two-tailed unpaired Student's t test. **E** Western blot assay for detecting the expression of β1-AR, β2-AR, β3-AR, and CCL2 in tumor tissue lysates from DEN + CCl₄-induced HCC model and subcutaneous Hepa1-6 tumors ($n = 4$ for each group). **F** Total tumor numbers (left) and numbers of tumor with diameter ø ≥ 3 mm (right) on livers from DEN + CCl₄-induced tumor-bearing mice. Mice were fed under SE or EE conditions with or without β-ARs blockade treatment (β-block, treated with SR59230A + propranolol). Vehicle+SE: $n = 7$, Vehicle+EE: $n = 9$, β-block+SE: $n = 9$, β-block+EE: $n = 9$. All data are presented as the mean ± SEM, and analyzed by two-tailed unpaired Student's t test. **G, H** Tumor volume of subcutaneous Hepa1-6 tumor-bearing mice **G** and H22 tumor-bearing mice **H** treated without (Vehicle, VEH) or with β-ARs blockade treatment (G, SE + VEH: $n = 8$, EE + VEH: $n = 12$, SE + β-block: $n = 10$, EE + β-block: $n = 12$; H, SE + VEH: $n = 8$, EE + VEH: $n = 8$, SE + β-block: $n = 11$, EE + β-block: $n = 8$). All data are presented as the mean ± SEM, and analyzed by two-tailed unpaired Student's t test with *$p < 0.05$, **$p < 0.01$, ***$p < 0.001$. **I** Tumor volume of subcutaneous Hepa1-6 tumors in mice injected with or without6-hydroxydopamine (6OHDA, 25 mg/kg) SE + VEH: $n = 12$, EE + VEH: $n = 11$, SE + 6OHDA: $n = 11$, EE + 6OHDA: $n = 14$. All data are presented as the mean ± SEM, and analyzed by two-tailed unpaired Student's t test with *$p < 0.05$, **$p < 0.01$, ***$p < 0.001$. **J-L** Percent of immune cells in tumors were detected by flow cytometry from Fig. 4G **H**, Fig. 4H **I** and Fig. 4I **J** (J, $n = 6$ for each group; K, $n = 6$ for SE + VEH and EE + VEH group, $n = 5$ for SE + β-block and EE + β-block group; L, $n = 10$ for SE + VEH and EE + VEH group, $n = 6$ for SE + 6OHDA and EE + 6OHDA group). All data are presented as the mean ± SEM, and analyzed by two-tailed unpaired Student's t test with *$p < 0.05$, **$p < 0.01$, ***$p < 0.001$.

tumor-bearing mice. The qRT-PCR results showed that EE housing led to a profound decrease of mRNA expression of CCL2 in both tumor cells and immune cells, including TAMs and G-MDSCs (Fig. 5G–I). Notably, these effects were subverted when β-ARs signaling was blocked (Fig. 5G–I). These results indicated that both the tumor-derived and host-derived CCL2 was regulated by β-ARs signaling and involved in EE-mediated tumor control.

To determine whether NE or EPI directly affect the CCL2 expression, we incubated them in vitro with human liver organoid, mouse tumor cells, mouse-derived BMDMs, and G-MDSCs (Supplementary Fig. 6E–H). Human hepatocytes/hepatic stellate cells organoid can better mimic the liver fibrosis process, which plays an essential role in hepatocarcinogenesis. Interestingly, we found that low concentrations of NE (50 μg/ml or 500 μg/ml) and EPI (50 μg/ml) reduced the expression level of CCL2 mRNA, while high dose of NE (5000 μg/ml) and EPI (500 μg/ml or 5000 μg/ml) increased CCL2 expression in human liver organoid (Supplementary Fig. 6E). Similarly, a U-shape relationship of NE/EPI concentration to CCL2 expression was also found in liver cancer cells (Supplementary Fig. 6 F), BMDMs (Supplementary Fig. 6G), and G-MDSCs (Supplementary Fig. 6H), suggesting that moderate NE/EPI level and activation of SNS/β-AR signaling led to better CCL2 control. In light of this, we further determined the relationship of NE/EPI concentration and tumor control. LPC-12 bearing mice under SE housing were administrated with different dose of NE or EPI, and tumor growth was monitored. The results showed that a low dose of 2 mg/kg NE/EPI inhibited s.c. tumor growth, but a high dose of 6 mg/kg EPI or 8 mg/kg NE promoted tumor growth in LPC-12 bearing mice (Supplementary Fig. 6I,J). These data suggest that the pro- or antitumor functions of EPI/NE depend on specific dosages in various situations and this would be interesting for further investigation.

Altogether, these data demonstrated the sympathetic modulation of tumor-derived and host CCL2 expression via β-ARs as a peripheral Neuro-Endocrine-Immune pathway for tumor regulation.

**EE overcomes PD-L1 based checkpoint blockade resistance.** Only small numbers of patients with HCC respond to PD-1-based immunotherapy[26]. PD-1-resistant patients usually present distinct signatures of upregulated genes involved in immunosuppression, angiogenesis, monocytes, and macrophages

chemotaxis[27,28]. As discussed previously, environmental eustress reshaped the immunosuppressive tumor microenvironment and induced an antitumor immunity, which drove a hypothesis that EE is likely to help overcome PD-1 resistance. To test this, we combined the EE housing and anti-PD-L1 treatment in mice bearing DEN + CCl₄-induced HCC tumor (Fig. 6A). In SE mice, DEN + CCl₄ treatment led to numerous tumor nodules in the liver, which were refractory to PD-L1 immunotherapy (Fig. 6B, C). Consistently, EE housing alone significantly reduced the tumor number and size (Fig. 6B, C). In stark contrast, mice in the combination groups showed a robust tumor control and a minimized tumor size (Fig. 6B, C). Additionally, EE housing significantly enhanced total (Fig. 6D) and IFN-γ producing tumor-filtrated CD8⁺ T cells (Fig. 6E) and elevated circulating IFN-γ level in mice treated with anti-PD-L1 antibody (Fig. 6F). Next, we recurred to transplantable LPC-H12 and H22 models to investigate the impact of EE on PD-L1 immunotherapy. C57BL/6 and BALB/c mice housing in SE or EE conditions were s.c. inoculated with LPC-H12 and H22 HCC tumor cells, respectively, followed by anti-PD-L1 treatment (Fig. 6G). In LPC-H12 tumor-bearing mice, the tumor growth was reduced in response to PD-L1 or EE housing alone, with 49.5 and 46.9% inhibition, respectively (Fig. 6H and Supplementary Fig. 7A). In comparison, a 72.8% tumor growth inhibition was observed in mice housing in EE and receiving PD-L1 treatment (Fig. 6H). H22 tumor was relatively refractory to PD-L1 treatment or EE housing alone, leading to growth inhibition of 25.9 and 36.6%, respectively (Fig. I and Supplementary Fig. 7A). Similarly, combination treatment had a synergistic effect on their antitumor actions with 62.3% of inhibition of tumor growth (Fig. 6I).

To further elucidate the mechanism of the synergy of EE housing to PD-L1 blockade, we investigated tumor-infiltrated immune cells in the tumor microenvironment by flow cytometry analysis and tracked antigen-specific T cells in the LPC-H12 tumor model. We found a significant increase of CD8⁺ T cells and a decrease of G-MDSCs and M2-like TAMs in the tumor tissues from EE mice, and PD-L1 treatment alone produced a similar but much less effective. Importantly, these effects were dramatically magnified in the combination group (Fig. 6J). We further tracked tumor antigen-specific T cells in the spleen of tumor-bearing mice. The results showed a synergistic increase of tumor-reactive IFN-γ producing T cells in EE mice with PD-L1 blockade (Fig. 6K). These data suggest that EE housing enhanced

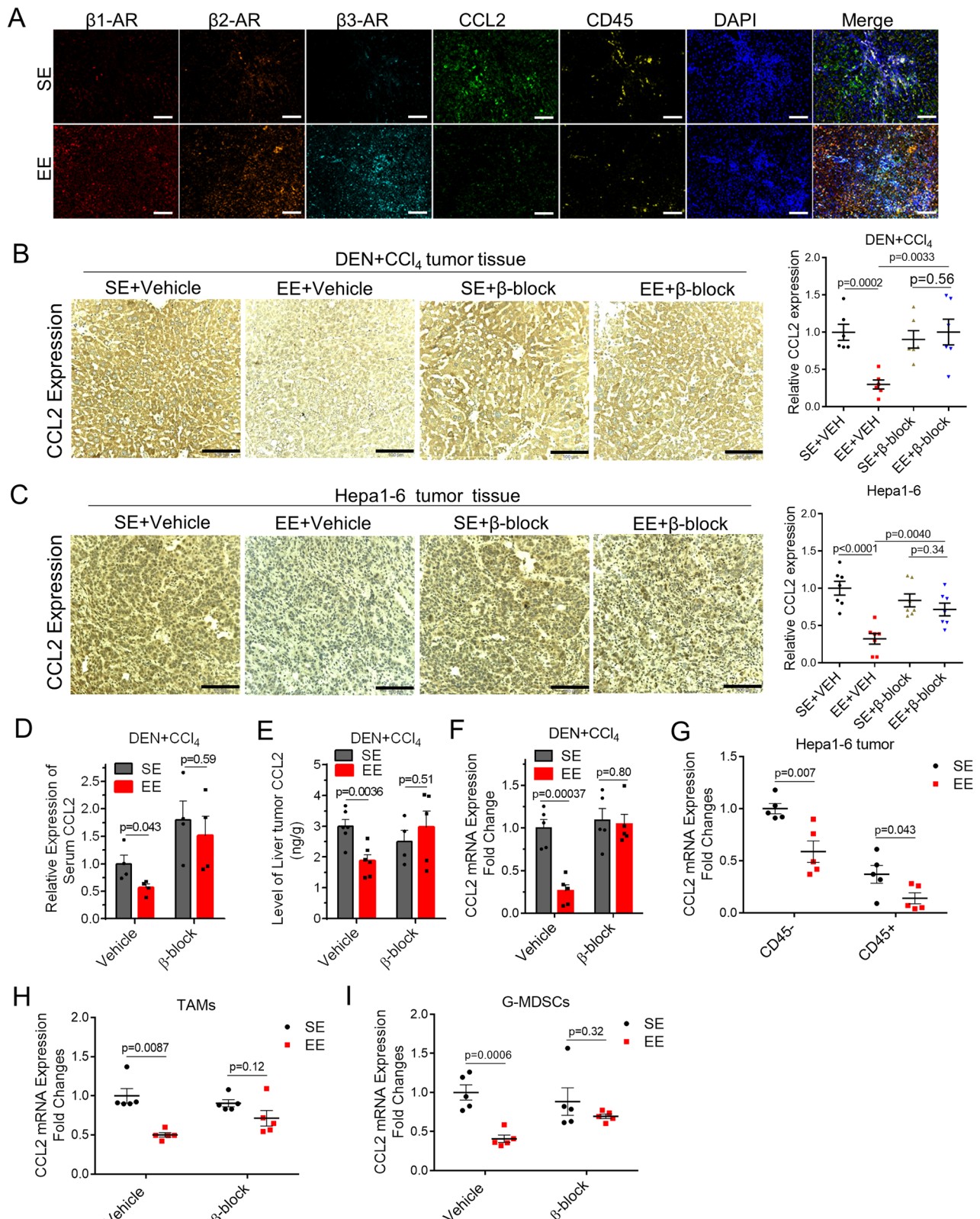

the tumor immunogenicity and increased CD8$^+$ T cell infiltration and tumor-specific CD8$^+$ T cell response when combined with PD-L1 blockade. These results demonstrated a potent synergy between the EE and PD-L1 blockade in control immunotherapy refractory HCC tumors.

Recent studies have shown the sympathetic and parasympathetic nerves in tumor tissues were correlated with PD-1/PD-L1[29].

As stated previously, EE-inhibited tumors via SNS/β-ARs, which drove a hypothesis that EE is likely to help overcome PD-L1 resistance through SNS/β-ARs. The data showed NE or EPI could mimic the inhibition of tumor and immunomodulatory effects of EE (Supplementary Fig. 5 K, L, and Supplementary Fig. 7B, C). Of note, mice in the NE or EPI combined with αPDL1 showed a robust tumor control and a minimized tumor size (Supplementary

**Fig. 5 Sympathetic modulation of CCL2 expression via β-ARs as a peripheral Neuro-Endocrine-Immune pathway for tumor regulation. A** Immunofluorescence staining of β1- Adrenergic Receptor (AR), β2-AR, β3-AR, CCL2, CD45, and DAPI in liver tumor tissues from DEN + CCl₄-induced tumor-bearing mice under SE or EE-feeding condition. Original magnification 20 × 10, Scale bar, 100 μm. **B** Immunohistochemical staining of CCL2 in the tumor tissue from DEN + CCl₄-induced tumor-bearing mice under Standard Environment (SE) and Enriched Enrichment (EE) feeding condition with or without β-AR blockade (β-block, treated with SR59230A + propranolol). The relative CCL2 expression was quantified by Image J analysis (n = 6 for each group). Original magnification 20 ×10, Scale bar, 100 μm. All data are presented as the mean ± SEM, and analyzed by two-tailed unpaired Student's t test. **C** Immunohistochemical staining of CCL2 in the tumor tissue from Hepa1-6 tumor-bearing mice under SE- or EE-feeding condition with or without β-AR blockade. The relative CCL2 expression was quantified by Image J analysis (n = 7 for each group). Original magnification 20 ×10, Scale bar, 100 μm. All data are presented as the mean ± SEM, and analyzed by two-tailed unpaired Student's t test. **D** Levels of CCL2 in serum from DEN + CCl₄-induced tumor-bearing mice under SE- or EE-feeding condition with or without β-AR blockade (n = 4 for each group). All data are presented as the mean ± SEM, and analyzed by two-tailed unpaired Student's t test. **E** ELISA analysis of CCL2 in the liver from DEN + CCl₄-induced tumor-bearing mice under SE or EE-feeding condition with or without β-AR blockade (Vehicle + SE: n = 6, Vehicle+EE: n = 6, β-block+SE: n = 4, β-block+EE: n = 5). All data are presented as the mean ± SEM, and analyzed by two-tailed unpaired Student's t test. **F** mRNA expression of CCL2 in the liver from DEN + CCl₄-induced tumor-bearing mice under SE- or EE-feeding condition with or without β-AR blockade (n = 5 for each group). All data are presented as the mean ± SEM, and analyzed by two-tailed unpaired Student's t test. **G–I** Mostly tumor cells (CD45⁻), immune cells (CD45⁺), TAMs cells (CD45⁺CD11b⁺F4/80⁺Ly6G⁻) and G-MDSCs (CD45⁺ CD11b⁺F4/80⁻Ly6G⁺) in the tumor microenvironment were sorted from subcutaneous Hepa1-6 tumor-bearing mice by flow cytometry after indicated treatments. mRNA expression of CCL2 were determined in tumor cells and immune cells **G**, TAMs **H)** and G-MDSCs **I**. n = 5 for each group. All data are presented as the mean ± SEM, and analyzed by two-tailed unpaired Student's t test.

Fig. 7 C). Blockade of β-AR signaling abolished the EE-mediated sensitization effect of PD-L1 based checkpoint blockade resistance in DEN + CCl₄-induced (Supplementary Fig. 7D, E) and LPC-H12 tumor models (Supplementary Fig. 7 F, G).

Altogether, these results indicate EE overcomes PD-L1-based checkpoint blockade resistance via the β-AR signaling pathway.

In patients with liver cancer, β-ARs is frequently down-regulated and is related to clinical prognosis.

To further demonstrate the role of β-ARs in HCC development and progression and to confirm its clinical relevance, we collected cohorts of HCC from The Cancer Genome Atlas (TCGA) data sets for which genome-wide gene expression and survival data were publicly available and examined how the β-ARs expression (i.e., ADRB1, ADRB2, and ADRB3) correlates with patient survival. The results showed that the expression of ADRB1 (Supplementary Fig. 8 A) and ADRB2 (Supplementary Fig. 8B) in HCC tumor tissue were downregulated compared to the TCGA normal tissue or adjacent-tumor tissues. Similarly to HCC, the expression of ADRB1 and ADRB2 were also significantly reduced across multiple types of human tumors such as breast cancer (BARC), cholangiocarcinoma (CHOL), colon cancer (COAD and READ), lung cancer (LUAD and LUSC, Supplementary Fig. 9 A, B). Interestingly, the gene expression of ADRB3 was minimal detected in most types of tumor and adjacent-tumor tissues, including HCC (Supplementary Fig. 8 C and Supplementary Fig. 9 C). We further used the Kaplan-Meier Plotter web server and examined the correlation of ADRB1, ADRB2, and ADRB3 expression levels to the prognosis of HCC patients. The results showed a positive association between β-ARs expression and patients Overall Survival (Supplementary Fig. 10 A) and Progression-Free Survival (Supplementary Fig. 10B). Under the fact that eustress-induced β-ARs expression and reduced CCL2 expression in mice, lower expression of CCL2 was correlated with favorable overall survival patients with HCC, as we previously reported[24].

## Discussion

Here we established a causal link between the environmental eustress and liver carcinogenesis and progression via subverting β-ARs/CCL2-mediated immunosuppressive effect. We demonstrated that mice living in EE showed better protection against carcinogen-induced liver neoplasia tumors and transplantable syngeneic liver tumors than those in SE, owing to a CD8⁺ T cell-dependent tumor control. Environmental eustress activated the peripheral SNS, moderately elevated plasma NE and EPI, and enhanced β-ARs signaling at the ligand and receptor levels. Activation of β-ARs in tumor cells and immune cells directly silenced the CCL2 expression, leading to an immunosuppressive tumor microenvironment and a significant enhancement of checkpoint blockade-refractory tumor response to anti-PD-L1 therapy (Fig. 6L).

The sympathetic nervous system (SNS) is commonly associated with the stress response, including distress and eustress. In response to stress and activation of SNS, NE, and EPI were upregulated and stimulated β-ARs locally and systematically, leading to both advantageous and harmful effects on organisms depending on the duration of the response and other unknown factors[13,30]. Studies attempting to inhibit tumor growth by wheel running in mice have shown that EPI injection daily with a low dose of 0.5 and 2 mg/kg for several days could inhibit s.c. tumor growth, mimicking the effect of Voluntary exercise[31]. While chronic treatment with β-agonist isoprenaline at a dose of 10 mg/kg daily promoted tumor development and impaired the antitumor immunity[32]. Our results showed that 0.5 and 2 mg/kg NE/EPI inhibited s.c. tumor growth but 6 mg/kg EPI or 8 mg/kg NE promoted tumor growth in mice (Supplementary Fig. 5 K, 6I–J, and 7B). Consistently, we found a U-shape relationship of NE/EPI concentration with CCL2 expression in cultured tumor cells, TAMs or MDSCs; that is, lower concentration of NE/EPI help to reduce the CCL2 expression, which higher concentration subverted this effect (Supplementary Fig. 6E–I). These data suggested that the functions of EPI/NE are dose-specific, which would be interesting for further investigation. Many studies have also produced a paradoxical relationship between β-AR activity and eustress versus distress, providing evidence that β-AR activation contributes to the distress-induced cancer-promoting effect[33–35], and has a tumor-protective role in eustress models[16,17,31]. The possible mechanisms might include the distinct patterns of tissue distribution of β-ARs (e.c. β1-AR, β2-AR, and β3-AR) and signal through distinct biochemical pathways, which functionally differ on cancer biology in a context-dependent and nonlinear manner.

Moreover, population studies revealed that severe depression appears to induce over-activation of β-ARs, whereas mild depression is associated with insufficient β-ARs activation[36–38], suggesting that moderate activation of β-ARs might be positive in distress relief. Here, we found that environment eustress moderately increased the level of NE and EPI in the blood serum and tumor microenvironment. Blockade of β-ARs, the receptors of NE and EPI, abrogated the tumor protective function of EE. Although more investigations would be required, this explains the

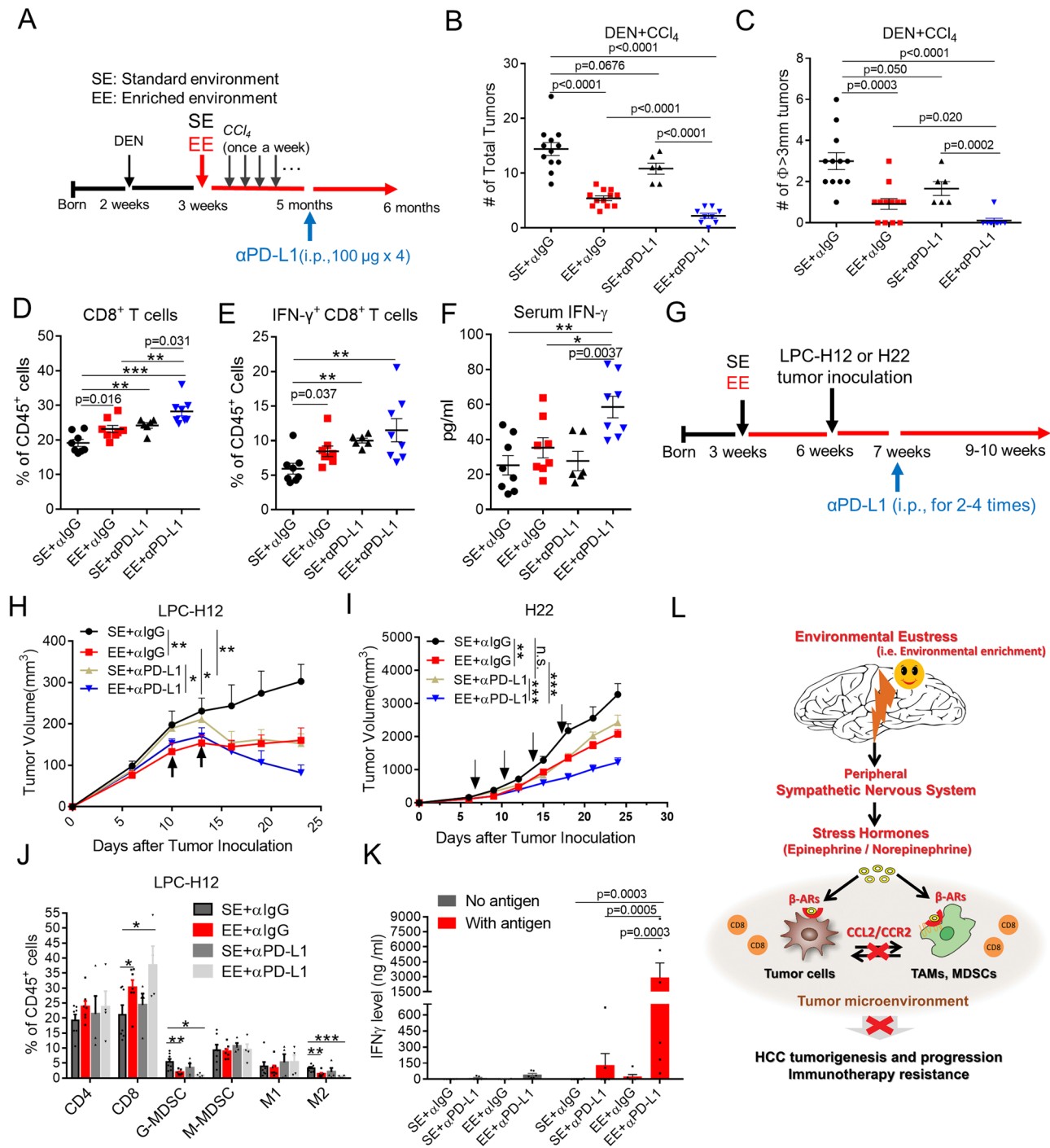

paradox function of β-ARs signaling in stress-modulated tumor growth to a certain extent. Of note, along with the moderate elevation of NE and EPI, the expression of β-ARs was significantly increased within the tumor microenvironment in mice housing in EE. Previous study also showed EE enhanced chronic activation of β-AR signaling at both the ligand and receptors level to prevent microglia inflammation by amyloid-β and provided protection against features of Alzheimer's disease[39]. Following this, we found that β-ARs were frequently downregulated and are related to the poor prognosis of patients with HCC.

We have reported that over-expression of CCL2 influenced hepatic TAMs and MDSCs accumulation and immunosuppressive features in liver cancer[24]. Subpopulations of TAMs and

MDSCs are both capable of suppressing CD8[+] T cells[40,41]. Previous studies showed therapeutic targeting on G-MDSC and Macrophages could enhance the effect of PD-1-based immunotherapy[41,42]. In this study, we found that EE could reduce serum CCL2, downregulate the expression of CCL2 in tumor cells and tumor-associated immune cells, and increase the number of CD8[+] T cells by suppressing TAMs and G-MDSCs infiltration. While more investigations would be needed, growing evidence suggests that immunosuppressive myeloid cells as a major cell type linking the stress and immune responses[13,43]. Moreover, some reports have revealed that CCL2 impairs cancer immunotherapy, and its blockade might thus be beneficial to the outcome of cancer therapy[44,45]. Here we found eustress overcame the PD-1 resistance via SNS/β-ARs signaling and increased the

**Fig. 6 Environmental eustress overcomes therapeutic PD-L1/PD-1 blockade resistance. A** Schematic representation of the experimental design and time line for DEN + CCl$_4$-induced tumor model with anti-PD-L1 immunotherapy, which includes one injection of DEN (25 mg/kg, i.p.) at 2 weeks old, multiple injections of CCl$_4$ (0.5 ml/kg, i.p., once a week) at 3 weeks old with SE or EE-feeding conditions, and PD-L1 treatment (100 μg/mice, twice a week for total 6 times) at 6 months. i.p.: intraperitoneal injection. **B, C** Total tumor numbers **B** and numbers of tumor with diameter ø ≥ 3 mm **C** in DEN/CCl$_4$-induced HCC mice treated with anti-PD-L1 immunotherapy under Standard Environment (SE) and Enriched Enrichment (EE) feeding conditions (SE + α IgG: $n = 12$, EE + αIgG: $n = 12$, SE + αPD-L1: $n = 6$, EE + αPD-L1: $n = 9$). All data are presented as the mean ± SEM, and analyzed by two-tailed unpaired Student's t test. **D, E** Total CD8$^+$ lymphocytes **D** and IFN-γ$^+$CD8$^+$ lymphocytes **E** from DEN + CCl$_4$-induced HCC tumor tissue analyzed by flow cytometry (SE + α IgG: $n = 8$, EE + αIgG: $n = 8$, SE + αPD-L1: $n = 6$, EE + αPD-L1: $n = 8$). All data are presented as the mean ± SEM, and analyzed by two-tailed unpaired Student's t test with *$p < 0.05$, **$p < 0.01$, ***$p < 0.001$. **F** The level of serum IFN-γ in mice bearing DEN + CCl$_4$-induced HCC tumor (SE + αIgG: $n = 8$, EE + αIgG: $n = 8$, SE + αPD-L1: $n = 6$, EE + αPD-L1: $n = 8$). All data are presented as the mean ± SEM, and analyzed by two-tailed unpaired Student's t test with *$p < 0.05$, **$p < 0.01$, ***$p < 0.001$. **G** Schematic representation was shown as the experimental design for subcutaneous LPC-H12 or H22 murine HCC tumor model with anti-PD-L1 immunotherapy under SE or EE-feeding conditions. **H, I** Tumor volume of C57BL/6 mice bearing subcutaneous LPC-H12 (H, SE + αIgG: $n = 7$, EE + αIgG: $n = 9$, SE + αPD-L1: $n = 9$, EE + αPD-L1: $n = 10$) or BALB/c mice bearing H22 tumor (I, SE + αIgG: $n = 11$, EE + αIgG: $n = 11$, SE + αPD-L1: $n = 9$, EE + αPD-L1: $n = 9$) treated with IgG or anti-PD-L1 antibody under SE or EE-feeding conditions. All data are presented as the mean ± SEM, and analyzed by two-tailed unpaired Student's t test with *$p < 0.05$, **$p < 0.01$, ***$p < 0.001$. **J** Infiltrated immune cells within LPC-H12 tumor microenvironment were analyzed by flow cytometry (SE + αIgG: $n = 7$, EE + αIgG: $n = 7$, SE + αPD-L1: $n = 4$, EE + αPD-L1: $n = 4$). All data are presented as the mean ± SEM, and analyzed by two-tailed unpaired Student's t test with *$p < 0.05$, **$p < 0.01$, ***$p < 0.001$. **K** The antigen-specific T cell activation was determined by the level of tumor-antigen induced IFN-γ secretion. Splenocytes isolated from LPC tumor-bearing C57BL/6 mice ($n = 6$) were stimulated with necrotic LPC-H12 cells via repetitive freeze-thaw. 48 h later, IFNγ production was detected by Cytometric Bead Array (CBA) mouse IFNγ assay. All data are presented as the mean ± SEM, and analyzed by two-way ANOVA. **L** Schematic mechanism of eustress-induced impairment of HCC tumorigenesis and progression via modulation of SNS/β-ARs/CCL2 axis and enhancement of tumor immunity.

tumor-specific CD8$^+$ T cell infiltration and reaction via silencing CCL2.

Overall, we identified a peripheral Neuro-Endocrine-Immune pathway SNS/β-ARs/CCL2 in the eustress model to relieve tumor immunosuppression and to overcome PD-1 immunotherapy resistance. In addition, we provided insights into how eustress functions in cancer prevention and control, ensuring better treatment outcomes of patients with liver cancer and more efficient therapeutic approaches.

## Methods

**Cell lines and reagents**. All murine liver cancer cell lines were purchased from ATCC, including Hepa1-6, LPC-H12, and H22. Human hepatocytes/hepatic stellate cells organoid were obtained from the Ding's Laboratory, SIBS, CAS. All cell lines were cultured in RPMI-1640 or DMEM medium (supplemented with 10% FBS and 1% penicillin/streptomycin) in a humidified incubator at 37 °C. The main reagents used in the study were shown in the Supplementary table 1.

**Animals**. C57BL/6 (male, 2–3 weeks age) and BALB/c mice (male, 3 weeks age) were obtained from Shanghai Slac Laboratory Animal Co. and fed in a pathogen-free vivarium under standard conditions (temperature, around 22 °C; relative humidity, 40–70%, and a 12-h-light–dark cycle). CCL2$^{-/-}$ and CCR2$^{-/-}$ mice on C57BL/6 background were obtained from the Jackson Laboratory. Animal protocols were performed according to the SIBS Guide for Care and Use of Laboratory Animals and approved by the Animal Care and Use Committee of Institute for Nutritional and Health, SIBS, CAS.

**Chemical-induced HCC models and transplantable HCC models**. For DEN + CCl$_4$ induced HCC model, the male C57/BL6 mice were injected intraperitoneally with diethylnitrosamine (DEN 25 mg/kg) at 14-day age and randomly assigned into EE and SE groups. At 4 weeks of age, for DEN combined with CCl$_4$ induced HCC model, the mice were administrated with CCl$_4$ (1 ml/kg, dissolved in olive oil), or olive oil alone (vehicle) and intraperitoneal injection once a week for up to the experiment ended; For DEN + HFD-induced HCC model, the mice were fed with HFD for up to 5–7 months. The HFD was 60% of kcal fat, 20% of kcal protein, and 20% of kcal carbohydrate. For single CCL4-induced HCC model, mice were injected intraperitoneally with CCl$_4$ (0.5 ml/kg) three times a week for up to 4–5 months.

To establish the transplantable subcutaneous HCC model, murine HCC cells (Hepa1-6, H22, and LPC-H12) were injected into the right flanks of the recipient mice with approximately $5 \times 10^5$–$1 \times 10^6$ cells. We recorded tumor occurrence by physical examination and measured the tumor size every 3 days starting from day 5 after inoculation. Tumor volumes were determined by caliper measurements in two perpendicular diameters of the implant and calculated using the formula $1/2a \times b^2$, where a is the long diameter and b is the short diameter.

**Environmental enrichment models**. Male 3-week-old C57/BL6 or BALB/c mice were housed in groups (8–12 mice per cage) in large cages (40 cm × 30 cm × 20 cm)

supplemented with running wheels, tunnels, huts, retreats, and wood toys in addition to standard lab chow and water. We housed control mice (four mice per cage) under standard laboratory conditions.

**Immune cells depletion, CCL2 neutralization, and PD-L1 immunotherapy**. For CD4$^+$ T cell, CD8$^+$ T cell, NK cell, and G-MDSCs depletion, mice were intraperitoneally injected with indicated antibodies (anti-CD4, anti-CD8, anti-NK1.1, anti-Ly6G) and the dose was 200 μg/mice, then followed by repeated injection every six days; for CCL2 neutralization, mice were intraperitoneally injected with 200 μg/mice anti-CCL2 antibody every three days. The depletion experiments above started one day before the injection of subcutaneous tumor. For macrophage depletion, the neutral Clodronate Liposomes was injected intraperitoneally (0.1 mL for 20–25 g body weight.) For PD-L1 treatment, H22 tumor-bearing mice were firstly administrated with 200 μg of PD-L1 blocking antibody on the 7th day of after tumor inoculation followed by another three times of 100 μg antibody every 3 days. LPC-H12 tumor-bearing mice were firstly given 100 μg of PD-L1 antibody on the 7th day of inoculation, and the second was given 3 days later at the same dose.

**Blockade of adrenergic receptors and SNS**. 4 week old of male C57BL/6 mice were randomized to drinking water containing 0.5 g/L β1/2-AR antagonist propranolol or nothing. β3-AR antagonist SR59230A (Sigma, 5 mg/kg) was intraperitoneally injected biweekly. α-AR antagonist phenoxybenzamine hydrochloride (Meilunbio, 10 mg/kg) was intraperitoneally injected once a week. Chemical sympathectomy (adrenergic denervation) was performed by intraperitoneal injection of 6OHDA (Sigma) at the dose of 25 mg/kg, every 3 days. Epinephrine or Norepinephrine (0.5 mg/kg) was intraperitoneally injected every 3 days.

**Flow Cytometry**. Fresh mouse tumor tissues were harvested, minced, and digested into single cell with mouse tumor dissociation kits (Miltenyi Biotech) according to the manufacturer's instructions. First, the single-cell suspensions were centrifuged and suspended in stain buffer (BD Pharmingen) after removal of red blood cells, and theninsuspended with Fixable viability stain 510 (BD Pharmingen, 1:500) to exclude the dead cells. Second, cells were incubated with the anti-mouse CD16/32 antibody (BD Pharmingen, 1:50) for 15 min to prevent non-specific binding. After this step, cells were stained with all relevant antibodies for 1 h at room temperature away from the light. Then cells were washed twice with PBS and re-suspended in 200 μL stain buffer. Last, single-cell suspensions were analyzed by BD FACS AriaII. According to isotype and fluorescence-minus-one (FMO), gating strategies were as follows: CD8$^+$ T cells (Live$^+$CD45$^+$CD3e$^+$CD8$^+$), CD4$^+$ T cells (Live$^+$CD45$^+$CD3e$^+$CD4$^+$), M1-TAMs (Live$^+$CD45$^+$Ly6G$^-$CD11b$^+$F4/80$^+$CD206$^-$), M2-TAMs (Live$^+$CD45$^+$Ly6G$^-$CD11b$^+$F4/80$^+$CD206$^+$), G-MDSCs (Live$^+$CD45$^+$CD11b$^+$Ly6G$^+$), M-MDSCs (Live$^+$CD45$^+$CD11b$^+$Ly6C$^+$), and NK cells (Live$^+$CD45$^+$CD3e$^-$NK1.1$^+$). The data were analyzed with FlowJo software, and gating strategy was shown in Supplementary Fig. 11, characterizing the immune cell infiltrates in tumor tissue. The FACs antibody used in the study was shown in the Supplementary table 2.

**Quantitative real-time PCR**. Total RNA was isolated from the indicated cells by use of the TRIzol reagent (Invitrogen) and reverse-transcribed into cDNA. Quantitative real-time PCR was performed on a 7900HT Fast Real-Time PCR System (Applied Biosystems) using SYBR green as the detection fluorophore. Target gene expression was normalized to the housekeeping genes, β-actin, and 18S. The primer sequences were provided in Supplementary Table 3.

**Western blotting**. Mouse tumor tissues were homogenized on ice in Protein extraction reagent buffer (BOSTER). For analysis of CCL2 protein expression in Hepa1-6 cells, cells were lysed in whole-cell lysate buffer (BOSTER). Total protein lysates were separated by gradient gel (10%, PAGE Gel Fast Preparation Kit, Epizyme), transferred to a PVDF membrane, and blotted overnight at 4 °C with the following primary antibodies, GAPDH (Cell Signaling, 1:1000), Anti-β1-AR (Absin, 1:1000), Anti-β2-AR (Absin, 1:1000), Anti-β3-AR (Absin, 1:1000), and Anti-CCL2 antibody (Abcam, 1:1000). Blots were rinsed and incubated with HRP-conjugate secondary antibody (Cell Signaling, 1:2000).

**Serum Epinephrine or Norepinephrine detection**. Mouse serum was obtained by allowing the blood to clot for 30 min on ice followed by centrifugation after euthanasia. Serum was at least diluted 1:5 in serum assay diluent and assayed using the Epinephrine/Norepinephrine ELISA Kit (Abnova) according to the manufacturer's protocols.

**Luminex Screening Assay analysis of mouse cytokines and chemokines**. The profiles of cytokine and chemokine levels in the mouse serums were measured using Mouse Magnetic Luminex Screening Assay 25-plex kits (R&D, LXSAMSM-25, USA) according to manufacturer's instructions. The analytes included CCL1, MCP-1(CCL2), CCL3, CCL4, CCL5, CLCX1, CXCL2, FGF-basic, G-CSF, GM-CSF, IFN- γ, IL-1α, IL-1β, IL-10, IL-12(p70), IL-13, IL-17A, IL-2, IL-3, IL-4, IL-5, IL-6, M-CSF, TNF-α, VEGF, MIP-1α, MIP-1β, RANTES, IP-10, and iNOS. Briefly, 50 μl of the serum from normal mice (non-tumor-bearing control mice) or tumor-bearing mice were incubated with fluorescently-dyed magnetic beads conjugated with antibodies against the indicated cytokines in single wells of 96-well microplates. Data from the reactions were acquired using a Bio-Plex 200suspension array system. The concentrations of 25 cytokines/chemokines were calculated with Bio-Plex Manager software 5.0 using a standard curve from a recombinant cytokine standard.

**Histology, Immunohistochemistry, and Immunofluorescence**. Mouse liver or liver tumor tissues were fixed overnight in 10% phosphate-buffered formaldehyde and embedded into paraffin block according to standard technical procedures. Histochemical, immunohistochemical, and immunofluorescent studies were performed on formalin-fixed and paraffin-embedded (FFPE) tissue sections. Sections were 3–6 μm thick for H&E or were 4 μm thick for immunohistochemical and immunofluorescent studies. In H&E, sections were deparaffinized with xylene and gradient alcohol, and were stained in Harris hematoxylin and the eosin-phloxine solution respectively. In immunohistochemical and immunofluorescent studies, the sections were dried in an oven in preparation for deparaffinization. For immunohistochemistry, slides were overlaid with citrate antigen retrieval solution (s1699, Dako, Hamburg, Germany) and incubated in saturated steam for 40 min. After cooling to room temperature, slides were washed 3 times with PBS, and incubated 30 min with 5% BSA before staining for reducing background staining. For staining, slides were incubated at 37 °C for 60 min with primary antibodies: anti-CD8, anti-F4/80, anti-Ly6C, anti-Ly6G, and anti-CCL2 (Abcam, 1:500), anti-β1-AR, anti-β2-AR, and anti-β3-AR (Absin, 1:1000) at the predetermined respective dilution and a HRP-conjugated Rabbit anti-rat IgG polyclonal (1:200) was used as the secondary antibody. Stained slides were counterstained with hematoxylin and analyzed using a Leica DM LB2 microscope, and images were captured with a Leica DFC 250 camera. Multiple fluorescence staining was obtained using PerkinElmer Opal kit (Perkinelmer). Antigen retrieval was performed in Target Antigen Retrieval Solution pH 9.0 (Dako) using microwave incubation. Afterwards, endogenous peroxidase was blocked and slides were washed with PBS. Slides were incubated in 5% BSA (RT, 30 min) for blocking reagent, primary antibodies (4 °C, overnight), secondary antibody (37 °C, 30 min), and TSA-Fluorescein (37 °C, 30 min). These steps were repeated until all primary antibodies were added. Staining used the following primary antibodies: CD45 (Abcam 1:800), CCL2 (Abcam 1:500), β1-AR, β2-AR, and β3-AR (Absin 1:1000). Immunofluorescence was imaged at 20X on the Vectra 3.0 Automated Quantitative Pathology Imaging System, and analyzed using in inForm Software 2.4.1 (all from Perkin-Elmer, Waltham, MA).

**Statistics and reproducibility**. Statistical analyses were performed with GraphPad Prism software using unpaired Student's t test (two-tailed) when two groups were being compared or two-way ANOVA when several groups were being compared. Similar results were obtained from three independent experiments. Results were depicted as means ± SEM at least two parallel assessments. The criterion for significance was set at a probability of less than 0.05. Adobe Photoshop CS6 and Microsoft PowerPoint 2016 was used to crop images from unprocessed images.

All experimental findings were replicated independently and reproducible with three times or more.

**Reporting Summary**. Further information on research design is available in the Nature Research Reporting Summary linked to this article.

## Data availability

The source data related to Figs. 1–6 and Supplementary Figs. 1–7 is provided as a Source Data file. The data in Supplementary Fig. 8 used in this study are available in the GEPIA web server database [http://gepia.cancer-pku.cn/detail.php?gene=&clicktag=boxplot]. The data in Supplementary Fig. 9 are available in the TIMER (Tumor IMmune Estimation Resource) database [http://timer.cistrome.org/]. The data in Supplementary Fig. 10 are available in the Kaplan-Meier Plotter database [http://kmplot.com/analysis/index.php?p=service&cancer=liver_rnaseq]. All the other data that support the findings of this study are available within the article, its Supplementary Information, or from the corresponding author upon reasonable request. A reporting summary for this article is available as a Supplementary Information file. Source data are provided with this paper.

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

## Acknowledgements

This study was supported by grants from the National Key R&D Program of China (2018YFC2000700), the National Nature Science Foundation of China (81630086, 81972820, 81672763), National Science and Technology Major Project (2017ZX09101002-002-005), the Key Research Program (ZDRW-ZS-2017-1) of the Chinese Academy of Sciences, the Major Science and Technology Innovation Program of Shanghai Municipal Education Commission (2019-01-07-00-01-E00059), Shanghai Pujiang Talent Program (19PJ1406900), Shanghai Young Eastern Scholar program, Shanghai Public Health System Construction Three-Year Action Plan (GWV-10.1-XK15), and Innovative research team of high-level local universities in Shanghai.

## Author contributions

C.L., Y.Y., C.C, L.L., X.L., and H.W. conceived and designed experiments. C.L., Y.Y., L.L., and X.L. analyzed data and wrote the manuscript. X.L. and H.W. supervised the project. C.L., Y.Y., C.C, L.L., and X.L. performed the in vitro and in vivo studies. Pathology analysis and tissue provision were accomplished by C.L., Y.Y., C.C., and L.L., L.Q., Q.C., J.L., X.W., and Q.B. assisted in experimental design and data evaluation. All authors reviewed and approved the manuscript.

## Competing interests

The authors declare no competing interests.
