## [Peer Review File · Nature Communications]

Reviewers' Comments:

Reviewer #1:

Remarks to the Author:

In the paper by Liu et al, the authors show that an increased level of catecholamines decrease CCL2 chemokine release by both tumor cells and immune cells. The authors also show that high levels of catecholamines from EE housing delays tumor growth. The decrease in the level of CCL-2 leads to CD8 infiltration and inhibits the accumulation of G-MDSC and M2 macrophages. The results are supported by different and multiple animal models. However, there are many questions raised and some significant concerns/questions which are listed below:

Major concerns:

1- It has been well documented in a variety of different papers that higher levels of catecholamines in serum promotes tumor growth. Here, authors conversely show that high level of catecholamines (EE) delays tumor growth. Is there a threshold in the level of catecholamines in which they could have pro or anti-tumor effects? There is insufficient experimental attention given to the discrepancy in results reported here and many other studies in the field.

2- Here authors show that EE housing which activates B-AR signaling increases the efficacy of anti-PD-1. This is also distinct from other findings, so it would be important for the authors to show that B2-AR agonists (to mimic EE housing) can improve the efficacy of anti-PD-1 too? Do B-blockers improve the efficacy of anti-PD1 in SE?

3- G-MDSC is one of the main immune cells affected by EE housing. But what about their function? Does EE housing affect immunosuppressive functions of MDSC?

Other questions/concerns:

4- The authors show that CCR2 deletion increases CD8 T cells infiltration into tumor but at same time decreases MDSC and TAM infiltration. Can authors explain how one receptor can have different effects of two different immune cells? Is CD8 infiltration indirectly mediated by low infiltration of TAMs and MDSCs? This may be included in the Discussion.

5- In Figure 2F, there is a difference in tumor growth between SE+ anti CD8 vs SE indicating that other immune cells such as NK cells may play a role in this model. This should be clarified.

6- It has been reported that norepinephrine increases CCL2 production by immune cells in both mouse and human cells (Takahashi et al, Burns 2004). Is the effect of Norepinephrine on CCL-2 expression different between immune cells isolated from healthy mice versus tumor bearing mice?

7- In Figure 3D the effect of CCR2 deletion is different among various myeloid cells. It would help if the authors show that CCR2 deletion increases CD8 T cells infiltration into tumor but at same time decreases MDSC and TAM infiltration. Can authors explain how one receptor can have different effects of two different immune cells? Is CD8 infiltration indirectly mediated by low infiltration of TAMs and MDSCs?

Reviewer #2:

Remarks to the Author:

Reviewer Comments to Authors

The work by Liu C et al, aims to study the influence of eustress on the development of liver cancer in mice. For that purpose, authors develop different HCC mouse models that were housed in Standard (SE) or an Enriched Environment (EE). Interestingly, authors describe that Environmental eustress reduces tumor growth and progression by remodeling the immune microenvironment and enhancing the CD8 T cell activity. Authors claim that immune microenvironment reshaping was dependent on CCL2/CCR2 signaling. Moreover, CCL2 expression was shown to be regulated by the sympathetic nervous system (SNS) via β -ARs. Finally, authors show some evidences of a synergy between EE and PD-L1 blockade, showing an increase in T cell

infiltration and tumor-specific T cell responses.

Although the manuscript does contain some interesting data, there are many deficiencies including lack of novelty in some of the aspects and important technical deficiencies in the methodology used, which are insufficient to support their conclusions. In general, this paper does not meet the quality that is characteristic of this journal and will modestly increase our understanding of the mechanisms behind the beneficial effects of environmental eustress on liver cancer herein.

Major comments

1. As mentioned by the authors in the introduction, the concept of the beneficial role of EE on the progression of some cancers (such as melanoma, colon cancer, breast cancer, pancreatic cancer and glioma), is not novel. Regarding the mechanism, in different tumors including glioma (PMID: 25818172) and pancreatic cancer (PMID: 28082402) it was demonstrated that the inhibitory effects on tumor growth were partly mediated through NK cell infiltration and NK cell-mediated cytotoxic effects. In agreement with this manuscript, in experiments carried out in animal models of pancreatic cancer it has also been previously shown that these antitumor responses were mediated via the sympathetic nervous system, as similar experiments carried out with beta-blockers or adrenergic nerve ablation with 6OHDA abolish the tumor-inhibitory effects of EE. However, in the manuscript by Liu C et al, authors claim that the mechanism of the tumor-growth inhibitory effects of EE in HCC are dependent on CD8+ T cells and CCL2/CCR2 axis. Nonetheless, authors mention that no appreciable changes are observed in the number of NK cells in the DEN+CCI4 as well as in transplanted syngeneic tumor models but this data are not even shown.
2. One of the main conclusions of this manuscript is that EE activates peripheral SNS and β -AR signaling both in tumor cells and tumor infiltrating immune cells, leading to silencing of CCL2 expression and activation of anti-tumor immunity. However, the strategies and methodologies used to properly conclude that this is the mechanism behind the protective effects of eustress on HCC tumor growth are not convincing. First of all, the strategy used to determine the expression of ARs in the tumor cells vs cells of the tumor microenvironment is not correct and tumor cell characterization is based on the CD45 negative expression, which is not entirely correct, as this classification might also include other cell types. Besides, one of their main conclusions regarding the EE-induced reduction of CCL2 levels is mainly based on the quantification of serum levels of this chemokine, which might not be necessarily a reflection of what happens in the liver but also a systemic effect. In this regard, the source of CCL2 could be indicative of the beneficial effects of EE in other organs or systems and not only the liver. Moreover, concerning this aspect CCL2 expression in the tumor tissue is only shown by immunohistochemistry in the DEN+CCL4 model (Figure 3), which is not even quantified and by qPCR in the CCL4-induced HCC and DEN+HFD HCC models, in which there are not significant differences (Suppl. Figure 3C,D). mRNA levels of Ccl2 expression by qPCR in the liver of mice with DEN-CCL4-induced liver carcinogenesis is not even shown.
3. The author's primary method of flow cytometry is very poorly described.
4. Although authors develop different HCC mouse models which are based both on carcinogen-induced and on transplantable syngeneic liver tumors, the potential mechanism of sympathetic modulation of CCL2 expression via β -ARs should be fully addressed on the setting of carcinogen-induced liver cancer models, as the effects of the tumor microenvironment in this condition is not properly reproduced in subcutaneous syngeneic mouse models. In this regard, there are many aspects of the mechanism that are only partly addressed in the DEN+CCI4 model. Once again, in one of the most important sections trying to unravel the mechanism of the sympathetic modulation of CCL2 expression via β -ARs, in which β -AR blockade is performed in DEN+CCI4 model to analyze CCL2 levels, the protein levels of this chemokine are shown in the serum and by immunofluorescence and immunohistochemistry (Figure 5). The quality of the immunofluorescence is very poor and immunohistochemistry images are not quantified (Figure 5 C,D).
5. Some of the statistical comparisons between the different groups are not clear enough to this reviewer. Some of the examples include Figure 2F, Figure 3H, Figure 4E,F,G, Figure 6H,I Suppl Figure 2C and Suppl Figure 3G. Authors should explicitly indicate which are the groups compared in each graph.
6. One of the main results in this study is that EE housing significantly reduces the DEN/CCI4-induced CCL2 levels. Although this is only verified in terms of serum levels and it should be confirmed at the hepatic level, being CCL2 a chemokine that is key for monocyte-derived macrophage recruitment to the liver, it would be interesting to measure the total counts of CCR2-

expressing macrophages in the liver after EE in the DEN/CCI4-induced HCC model compared to mice that have been housed in a standard environment.

7. The Discussion of the manuscript should highlight the relevance of this study, contextualizing their work according to the recent literature in this field, instead of enumerating or summarizing the results again.

Minor comments

- Regarding the HCC mouse models used in this manuscript chronic CCL4 administration is not a proper model of liver carcinogenesis.
- All the IHC images should include scale bars and should be quantified.
- In the experiments carried out with syngeneic transplantable tumors, besides showing the progression of tumor growth by measuring the tumor volume, authors should also include the final tumor weight as shown in Figure 1 (I, L), Figure 3 (F,H) and Suppl Figure 4(G,H).
- Fig 2D and 2E are not mentioned in the main text.
- In Figure 3B authors should quantify the expression levels of CCL2 by Western Blotting and additionally show the mRNA levels of this chemokine by qPCR. Similarly, regarding Suppl. Figure 3F, the results of the CCL2 protein levels shown by immunoblotting should also be confirmed by qPCR.
- In Figure 4C, the protein levels of β -ARs should be assessed by western blotting.
- In Figure 5A, authors claim "the extensive expression of β 1-AR and β 3-AR in both tumor cells and immune cells, and β 2-AR commonly expressed on immune cells". However, in Figure 5A authors do not include any tumor specific marker in the immunofluorescence assay and therefore, they cannot affirm this.
- In Figure 6E, authors should confirm the results taking out the possible outlier in the EE+aPD-L1 group.
- The manuscript should be thoroughly revised to correct typos and grammar mistakes.

Reviewer #3:

Remarks to the Author:

In this manuscript, the authors demonstrate that EE inhibits the growth of carcinogen-induced liver neoplasias and transplantable syngeneic liver tumors. They show that EE activated peripheral SNS and β -ARs signaling in tumor cells and tumor infiltrated myeloid cells, leading to silencing of CCL2 expression and activation of anti-tumor immunity.

Overall, this is an important paper but at this point, the manuscript contains some overstatements, unclear concepts, unclear data presentation, editing and statistics issues, and, in general, it feels more like a collection of stories.

- The authors use M1 and M2 definition that has been challenged in recent years and maybe better to use the specific, functional, cell characterization. Also, the gating for myeloid cells is not clear. It appears as if Ly6G was used twice in the gating (Fig s9)
- What is the difference between graphs 2B and 3G, 4J in terms of the effects on M1 (the graphs don't seem to demonstrate the same effect)?
- Please show the data for the NK cells. Especially since in previous publications NK was shown to play an important role.
- Please add a magnified image of the CCL2 staining (3B). In general, most of IH staining should be quantified.
- Another method of CCL2 quantification in the tumor will be useful (even mRNA)
- Fig 4: The authors argue that the NE effect is local but this is not shown directly. It is possible for example that the effect is mediated via changes in the bone marrow innervation.
- 4H, why didn't the author made the distinction between M1 and M2
- 5A- the staining is not clear at all. What is the condition (EE)? What is the comparison?
- 5E- the definition of tumor cells as CD45-. It does not make sense. These cells can be fibroblasts, for example.
- Do the levels of CCR2 change?
- What happens to CCL2 in CD8 cells in the EE?
- Supp. Fig 2. – it is not clear what the images indicates

- Many supplementary graphs are not clear (for example, Fig s3G). 3C, D – is missing statistics.
- In general, it may be useful to perform a statistical review of the paper
- The authors refer to norepinephrine and epinephrine as hormones, but they are not classically considered hormones, so this point requires clarification.
- The connection of PDL-1 should be explained in more detail.
- Many studies show that beta-blockers are in fact protective against tumors, therefore the argument that it's the opposite, requires more careful discussion and integration with the existing literature.
- In addition, many studies refer to b2 receptor in the context of the tumor but here, the authors link their work mainly the b3 and b1 receptor. This gap should be discussed.
- The tumor models are not described in sufficient detail.
- To what extent the effects is the EE and not just physical activity?
- The authors indicate that EE attentats growth, but it can also be reduced insemination.

Response letter

REVIEWER COMMENTS

Reviewer #1 (Remarks to the Author): with expertise in stress and cancer immunology/immunotherapy

In the paper by Liu et al, the authors show that an increased level of catecholamines decrease CCL2 chemokine release by both tumor cells and immune cells. The authors also show that high levels of catecholamines from EE housing delays tumor growth. The decrease in the level of CCL2 leads to CD8 infiltration and inhibits the accumulation of G-MDSC and M2 macrophages. The results are supported by different and multiple animal models. However, there are many questions raised and some significant concerns/questions which are listed below:

Major concerns:

1. It has been well documented in a variety of different papers that higher levels of catecholamines in serum promotes tumor growth. Here, authors conversely show that high level of catecholamines (EE) delays tumor growth. Is there a threshold in the level of catecholamines in which they could have pro or anti-tumor effects? There is insufficient experimental attention given to the discrepancy in results reported here and many other studies in the field.

Response: Thanks for the comment. The Sympathetic nervous system (SNS) is commonly associated with the stress response, including distress and eustress. In response to stress and activation of SNS, NE and EPI were upregulated and stimulated β -ARs locally and systematically, leading to both advantageous and harmful effects on organisms depending on the duration of the response and other unknown factors[1, 2].

According to current research literatures, a paradoxical relationship is found between β -ARs activity and tumor control in the context of eustress or distress models. It seems that β -ARs activation contributed to the cancer-promoting effect of distress[3-5], whereas required in tumor-protective effect of eustress models[6-8]. Moreover, voluntary running, one of the major components in EE, dramatically reduced the melanoma growth and metastasis in a Epinephrine dependent manner[9]. Blockade of β -ARs signaling blunts the exercise-induced tumor suppression[9]. These results indicated the activation of β -ARs might functionally differ on cancer biology in a context-dependent and non-linear manner.

In this study, we showed that EE housing could increase the level of NE and EPI in the blood serum and tumor tissues in tumor bearing mice (Fig.4B). We further demonstrated that EE activated peripheral SNS and β -ARs signaling in tumor cells and tumor infiltrated myeloid cells, leading to silencing of CCL2 expression and activation of anti-tumor immunity. Here, we found a U-shape relationship of NE/EPI concentration with CCL2 expression in human liver organoid, cultured tumor cells, TAMs or MDSCs and tumor control, that is, lower concentration of

NE/EPI help to reduce the CCL2 expression, while higher concentration subverted this effect (Supporting Fig1.A-E and also seen in Supplementary Fig.6E-H). Previous studies on the antitumor effect of running in mice have shown that EPI injection daily with a low-dose of 0.5mg/kg for several days could inhibit s.c. tumor growth, mimicking the effect of Voluntary exercise[8]. While chronic treatment with β -agonist isoprenaline at a dose of 10mg/kg daily promoted tumor development and impaired the antitumor -immunity [10]. **Our results showed that 0.5mg/kg and 2mg/kg NE/EPI inhibited s.c. tumor growth, but 6mg/kg EPI or 8mg/kg NE promoted tumor growth in mice (Supporting Fig1.F, G and Supplementary Fig.5K and Fig.6I,J).** These data suggest that the pro- or anti-tumor functions of EPI/NE depend on specific dosages in various situations and this would be interesting for further investigation. **These results have been added to the revised manuscript. Please see detailed changes in Supplementary Fig. 6.**

Supporting Fig.1 The U-shape relationship between NE/EPI concentration, CCL2 expression, and tumor growth.

(A) Human hepatocytes/hepatic stellate cells organoids were treated with varied concentration of NE and EPI (ng/ml) in vitro for 24h, followed by washing and medium replacement. 48 h later, CCL2 mRNA expression was determined with qPCR assay with β -actin as an internal control (n=3).

(B, C) The mRNA expression of CCL2 was determined with qPCR assay in Hepa1-6 (F) and

LPC-H12 (G) cells after a 24h-treatment with varied concentration of NE and EPI (50 or 500 ng/ml) in vitro (n=3).

(D) Bone marrow derived macrophages (BMDMs) were exposed to conditioned medium (CM) of Hepa1-6 tumor cells for 48h and subsequently treated with vehicle, NE and EPI (0.5, 5, 50, 500 ng/ml) in vitro for 24h. Cells were washed followed by medium replacement. 48h later, the mRNA expressions of CCL2 in BMDM were determined

(H) Bone marrow cells were isolated from normal C57BL/6 mice and cultured in the presence of recombinant murine granulocyte macrophage colony-stimulating factor (GM-CSF) for up to 7 days. Ly6G⁺ G-MDSCs were sorted out and subsequently treated with vehicle, NE and EPI (0.5, 5, 50, 500 ng/ml) in vitro. The mRNA expressions of CCL2 in G-MDSCs were determined after treatment (n=3).

(F, G) Tumor volume of subcutaneous LPC-H12 tumors in mice injected s.c. with different doses of NE or EPI every three days under SE feeding conditions (n=4-6).

2. Here authors show that EE housing which activates B-AR signaling increases the efficacy of anti-PD-1. This is also distinct from other findings, so it would be important for the authors to show that B2-AR agonists (to mimic EE housing) can improve the efficacy of anti-PD-1 too? Do B-blockers improve the efficacy of anti-PD1 in SE?

Response: Thanks for the comment. Recent studies have shown the sympathetic and parasympathetic nerves in tumor tissues were correlated with the expression of PD-1/PD-L1[11]. As previously discussed in the paper, EE inhibited tumors via SNS/ β -ARs, which drove a hypothesis that EE is likely to help overcome PD-L1 resistance through SNS/ β -ARs. Indeed, we demonstrated that EE housing increases the efficacy of anti-PD-1 via activating β -AR signaling. As suggested, we further tested the effect of β -AR

agonists on therapeutic efficacy of anti-PD-1. We showed a U-shape relationship between NE/EPI concentration (as β -ARs agonists) and tumor growth (**Supporting Fig.1F,G**). Our data showed that low dose of NE or EPI (2 mg/kg) could mimic the anti-tumor and immunomodulatory effects of EE (**Supporting Fig.2B**). **Of note, mice in the NE+ α PD-L1 or EPI+ α PD-L1 groups showed a robust tumor control indicated by the smallest tumor size, suggesting that β -ARs agonists could improve the efficacy of anti-PD-1 (Supporting Fig.2B). Blockade of β -AR signaling abolished the effect of EE-mediated overcome anti-PD-L1 resistance in DEN+CCl₄ (Supporting Fig.2D) and LPC-H12 tumor models (Supporting Fig.2F), but had no obvious effect on the therapeutic effect of anti-PD-L1 in SE mice (Supporting Fig.2F). Altogether, these results confirmed that EE overcomes anti-PD-L1 resistance via modulating β -ARs signalings.**

These results have been added to the revised manuscript. Please see detailed changes in Supplementary Fig. 7B-G.

Supporting Fig. 2 (Also seen in Supplementary Fig. 7) Environmental eustress enhances tumor β -AR signaling to augment the therapeutic of PD-L1/PD-1 blockade.

(A) Scheme of experimental procedure for subcutaneous LPC-H12 tumor model with NE or EPI treatment (2mg/kg) combined with anti-PD-L1 immunotherapy.

(B) Tumor volume and tumor weight of C57BL/6 mice bearing subcutaneous LPC-H12 tumors with NE or EPI treatment plus anti-PD-L1 immunotherapy (n=6).

(C) Scheme of experimental procedure for DEN+CCl₄-induced tumor model with or without anti-PD-L1 immunotherapy, or/and β -ARs blockade treatment (β -block: SR59230A+propranolol).

(D) Total tumor numbers (left) and numbers of tumor with diameter $\phi \geq 3$ mm (right) on livers from DEN+CCl₄-induced tumor-bearing mice with or without anti-PD-L1 immunotherapy, or/and β -ARs blockade treatment (n=4-10).

(E) Scheme of experimental procedure for subcutaneous LPC-H12 tumor model treated with or without anti-PD-L1 immunotherapy, or/and β -ARs blockade.

(F) Tumor volume and tumor weight of C57BL/6 mice bearing subcutaneous LPC-H12 tumors treated with or without anti-PD-L1 immunotherapy, or/and β -ARs blockade (n=10-12).

All data are presented as the mean \pm SEM from three independent experiments, and analyzed by two-way ANOVA with n.s., $p > 0.05$, *, $p < 0.05$; **, $p < 0.01$; ***, $p < 0.001$.

3. G-MDSC is one of the main immune cells affected by EE housing. But what about their function? Does EE housing affect immunosuppressive functions of MDSC?

4. The authors show that CCR2 deletion increases CD8 T cells infiltration into tumor but at same time decreases MDSC and TAM infiltration. Can authors explain how one receptor can have different effects of two different immune cells? Is CD8 infiltration indirectly mediated by low infiltration of TAMs and MDSCs? This may be included in the Discussion.

7. In Figure 3D the effect of CCR2 deletion is different among various myeloid cells. It would help if the authors show that CCR2 deletion increases CD8 T cells infiltration into tumor but at same time decreases MDSC and TAM infiltration. Can authors explain how one receptor can have different effects of two different immune cells? Is CD8 infiltration indirectly mediated by low infiltration of TAMs and MDSCs?

Response: Thanks for the comments. For a better response, we grouped three questions together. TAMs and MDSCs are known as major immunosuppressive cell subsets to suppress CD8⁺ T cells infiltration and their anti-tumor function [12, 13]. Previous studies have shown that therapeutic targeting of G-MDSC and Macrophages could enhance the effect of PD-1 based immunotherapy[13, 14]. In EE mice, qRT-PCR analysis of Hepa1-6 tumors showed a significant increase of pro-inflammatory (IL-12a, iNOS, IL-6, TNF, CD14, IFN-gamma, IFN-beta) and a dramatic decrease of anti-inflammatory (Arg1, mMGL2, Fizz1, CD163, Retn1a, IL-10, TGF-beta) compared to those in SE mice (**Supporting Fig.3A**). These results suggested that EE boosted inflammation in the tumor microenvironment. To further explore the influence of EE on CD8⁺ T cells through regulating G-MDSCs and macrophages, primary CD8⁺ T cells were cocultured with polarized G-MDSCs or macrophages *in vitro* which isolated from femurs of wild type mice or from the subcutaneous layer of Hepa1-6 tumor-burden mice under either SE or EE feeding condition (**Supporting Fig. 3B-E**). **Our results showed that the CD8⁺ T cells proliferated more when cocultured with G-MDSCs or macrophages from the EE mice compared to that from the SE mice, and strikingly, G-MDSCs can no longer inhibit CD8⁺ T cell proliferation in tumor bearing mice under EE conditions (Supporting Fig. 3B-E).**

To further determine the role of CCL2/CCR2 signaling on EE-mediated anti-tumor immunity, we analyzed the DEN+CCl₄-induced tumorigenesis in WT and CCR2 KO mice and investigated tumor-infiltrated immune cells by flow cytometry analysis. In SE mice, CCR2 deletion reduced the tumorigenesis, and increased CD8 T cells infiltration into tumor and decreases MDSC and TAM infiltration, which partially mimicked the effect of EE housing (**Fig. 3C,D**). Meanwhile, the protective effect of EE against DEN/CCl₄-induced hepatocarcinogenesis was disappeared in CCR2^{-/-} mice (**Fig. 3C,D**). Similarly, blockade of CCL2/CCR2 signaling with an anti-CCL2

antibody also abrogated the EE induced inhibitory effect on tumor growth in DEN+CC14 (**Fig.3E**) and Hepa1-6 models (**Fig.3F**). Flow cytometry analysis further revealed that blocking CCL2 in SE mice mimicked the effect of EE-mediated reshaping of the tumor microenvironment, that is, increase of CD8⁺ T cells and M1-TAMs and decrease of G-MDSCs and M2-TAM (**Fig.3G**). As suggested, to further determine the relationship between TAMs/G-MDSCs and tumor infiltrated CD8 T cells, we depleted the TAMs and G-MDSCs with Liposome and anti-Ly6G Ab in subcutaneous Hepa1-6 tumor model, respectively. **In SE mice, TAMs or G-MDSCs depletion induced more tumor infiltrated CD8⁺ T cells, which mimicked the effect of EE housing (Supporting Fig. 3F). Moreover, EE housing could not further increase the tumor-infiltrated CD8⁺ T cells when depleting TAMs or G-MDSCs (Supporting Fig. 3F). Taken together, these results suggest that EE increases CD8⁺ T cells infiltration by suppressing the immunosuppressive role of TAMs and G-MDSCs dependent on CCL2/CCR2 signaling.**

These results have been added to the revised manuscript. Please see detailed changes in Supplementary Fig.3

Supporting Fig.3 (Also seen in Supplementary Fig.3) Environmental eustress abolishes the immunosuppressive effect of G-MDSC and M2 macrophages on CD8⁺ T cells.

(A) mRNA expressions of typical pro-inflammatory and anti-inflammatory markers were quantified by qRT-PCR assay in subcutaneous Hepa1-6 tumors under SE or EE feeding condition (n=5).

(B-E) Bone marrow-derived cells were isolated from the femurs of normal mice or subcutaneous Hepa1-6 tumor-burden mice under SE or EE feeding condition. Bone marrow-derived cells were further polarized into G-MDSC by GM-CSF (B-C) or M2 macrophages by M-CSF and IL-4/IL-13 treatment (D-E). CD8⁺ T cells isolated from normal mice spleen were labeled with CFSE and cocultured with G-MDSC or M2 macrophages for 48h with the stimulation of IL-2, anti-CD3, and anti-CD28 antibody. The proliferation rates of CD8⁺ T cells were detected by flow cytometry (n=6). (F) The proportion of CD8⁺ T cells were

analyzed by flow cytometry in subcutaneous Hepa1-6 tumor microenvironment with macrophages or G-MDSC depletion from Fig.S2D and E models (n=5).

All data are presented as the mean \pm SEM from three independent experiments, and analyzed by unpaired Student's t-test or two-way ANOVA with n.s., $p > 0.05$; *, $p < 0.05$; **, $p < 0.01$; ***, $p < 0.001$.

5. In Figure 2F, there is a difference in tumor growth between SE+ anti CD8 vs SE indicating that other immune cells such as NK cells may play a role in this model. This should be clarified.

Response: Thanks for the comment. In this study, we found that CD8⁺ T cells depletion abolished the tumor-suppressive effect of EE in Hepa1-6 (Fig.2F), H22 (Fig.2G) and LPC-H12 (Fig.2H) tumor models, and similar results were found in DEN+CCl₄-induced HCC mouse model (Supporting Fig5.A and also seen in Fig.2I). Flow cytometry analysis showed that NK cells had no changes in cell proportion in SE and EE groups (Supporting Fig4.B, C). Even mice were treated with anti-NK1.1 depletion antibodies in conjunction with either SE or EE housing, EE is still the main factor that promotes the tumor growth after NK cell depletion (Supporting Fig4.D and also seen in Supplementary Fig.2F). Moreover, previous studies have reported that NK cells alone could not change tumor growth effectively in HCC mice models[15].

Taken together, NK cells certainly play important roles in the immune response, however, in our study design, these cells may not be the main contributors to the tumor growth.

These results have been added to the revised manuscript. Please see detailed changes in Supplementary Fig.2F-H.

Supporting Fig.4 Environmental eustress reshapes tumor microenvironment and reduces tumor growth dependent on CD8⁺ T cells, but not NK cells.

(A) Total tumor number and diameter $\varnothing \geq 3\text{mm}$ tumor number in the liver of DEN+CCl₄-treated mice with CD8⁺T cells depletion under SE or EE feeding conditions. (n=5-10)

(B-C) The proportion of NK cells were analyzed by flow cytometry in subcutaneous Hepa1-6 tumor microenvironment(B and C) and DEN+CCl₄-induced HCC microenvironment (C) (n=6).

(D) Tumor volume and tumor weight of subcutaneous Hepa1-6 tumors in C57BL/6 mice with NK cells depletion (treated with anti-NK1.1 neutralization antibody, n=8-11).

All data are presented as the mean \pm SEM from three independent experiments, and analyzed by unpaired Student's t test or two-way ANOVA with n.s., $p > 0.05$; *, $p < 0.05$; **, $p < 0.01$; ***, $p < 0.001$.

6. It has been reported that norepinephrine increases CCL2 production by immune cells in both mouse and human cells (Takahashi et al, Burns 2004). Is the effect of Norepinephrine on CCL-2 expression different between immune cells isolated from healthy mice versus tumor bearing mice?

Response: As supporting Fig.1 shown, we found a U-shape relationship of NE/EPI concentration with CCL2 expression in human liver organoid, cultured tumor cells, TAMs or MDSCs, and low-dose of NE/EPI inhibited s.c. tumor growth, but high-dose promoted tumor growth in mice (also seen in supporting Fig.5A,B). We found that low doses of NE/EPI reduced serum CCL2 in normal and tumor-bearing mice to a similar level (Supporting Fig.5C). To compare the effect of EPI and NE on CCL2 secreted by immune cells in wild type mice and tumor-bearing mice, we isolated BMDMs and G-MDSCs from bone marrow and treated them with different doses of EPI and NE. The results showed that no difference in G-MDSC concentration between wild-type and tumor-bearing mice treated for all different doses of NE or EPI tested. But the U-shape relationship between CCL2 concentration and NE/EPI doses was retained. (Supporting Fig.5D). However, when treated with higher-doses of NE or EPI, BMDMs from tumor-bearing mice had higher secretion levels of CCL2 than the wild type mice (Supporting Fig.5E).

Supporting Fig.5 Difference of CCL2 secretion of BMDMs and G-MDSCs from tumor-bearing mice and normal mice treated with various doses of NE and EPI.

(A, B) Tumor volume of subcutaneous LPC-H12 tumors in mice injected s.c. with different doses of NE or EPI every three days under SE feeding conditions (n=4-6).

(C) ELISA analysis of CCL2 in serum from normal mice or LPC-H12 tumor-bearing mice which injected s.c. with 2mg/kg of NE or EPI every three days under SE feeding conditions.

(D-E) Bone marrow cells were isolated from normal C57BL/6 mice and LPC-H12 tumor-bearing mice and cultured in the presence of recombinant murine granulocyte macrophage colony-stimulating factor (GM-CSF) for up to 7 days. (A) Ly6G⁺ G-MDSCs were sorted out and subsequently treated with vehicle, NE and EPI (0.5, 5, 50, 500 ng/ml) in vitro for 24h. (B) Bone marrow derived macrophages (BMDMs) were treated with vehicle, NE and EPI (0.5, 5, 50, 500 ng/ml) in vitro for 24h. The CCL2 in supernatant was determined after treatment using ELISA assay (n=3).

Reviewer #2 (Remarks to the Author): with expertise in liver cancer - mouse models

Reviewer Comments to Authors

The work by Liu C et al, aims to study the influence of eustress on the development of liver cancer in mice. For that purpose, authors develop different HCC mouse models that were housed in Standard (SE) or an Enriched Environment (EE). Interestingly, authors describe that Environmental eustress reduces tumor growth and progression by remodeling the immune microenvironment and enhancing the CD8 T cell activity. Authors claim that immune microenvironment reshaping was dependent on CCL2/CCR2 signaling. Moreover, CCL2 expression was shown to be regulated by the sympathetic nervous system (SNS) via β -ARs. Finally, authors show some evidences of a synergy between EE and PD-L1 blockade, showing an increase in T cell infiltration and tumor-specific T cell responses.

Although the manuscript does contain some interesting data, there are many deficiencies including lack of novelty in some of the aspects and important technical deficiencies in the methodology used, which are insufficient to support their conclusions. In general, this paper does not meet the quality that is characteristic of this journal and will modestly increase our understanding of the mechanisms behind the beneficial effects of environmental eustress on liver cancer herein.

Major comments

1. As mentioned by the authors in the introduction, the concept of the beneficial role of EE on the progression of some cancers (such as melanoma, colon cancer, breast cancer, pancreatic cancer and glioma), is not novel. Regarding the mechanism, in different tumors including glioma (PMID: 25818172) and pancreatic cancer (PMID: 28082402) it was demonstrated that the inhibitory effects on tumor growth were partly mediated through NK cell infiltration and NK cell-mediated cytotoxic effects. In agreement with this manuscript, in experiments carried out in animal models of pancreatic cancer it has also been previously shown that these anti-tumor responses were mediated via the sympathetic nervous system, as similar experiments carried out with beta-blockers or adrenergic nerve ablation with 6OHDA abolish the tumor-inhibitory effects of EE. However, in the manuscript by Liu C et al, authors claim that the mechanism of the tumor-growth inhibitory effects of EE in HCC are dependent on CD8⁺T cells and CCL2/CCR2 axis. Nonetheless, authors mention that no appreciable changes are observed in the number of NK cells in the DEN+CCI4 as well as in transplanted syngeneic tumor models but this data are not even shown.

Response: Thanks for the comment. In this study, we found that CD8⁺ T cells depletion abolished the tumor protective effect of EE in Hepa1-6 (**Fig.2F**), H22 (**Fig.2G**) and LPC-H12 (**Fig.2H**) tumor models, and similar results showed in

DEN+CCl₄ -induced HCC mouse model (Supporting Fig.6A and also seen in Fig.2I). Flow cytometry analysis showed NK cells had no change in SE and EE groups (Supporting Fig. 6B, C). Mice were treated with anti-NK1.1 depletion antibodies in conjunction with SE or EE housing, while EE still controlled tumor growth after NK cell depletion (Supporting Fig.6D and also seen in Supplementary Fig.2F). Studies have reported that given NK cells alone could not change tumors effectively in mice HCC models[15].

Taken together, NK cells certainly play important roles in the immune response, however, in our study design, these cells may not be the main contributors to the tumor growth.

These results have been added to the revised manuscript. Please see detailed changes in Supplementary Fig.2F-H.

Supporting Fig.6 Environmental eustress reshapes tumor microenvironment and reduces tumor growth dependent on CD8⁺ T cells, but not NK cells.

(A) Total tumor number and diameter ≥ 3 mm tumor number in the liver of DEN+CCl₄-treated mice with CD8⁺T cells depletion under SE or EE feeding conditions. (n=5-10)

(B-C) The proportion of NK cells were analyzed by flow cytometry in subcutaneous Hepa1-6 tumor microenvironment(B and C) and DEN+CCl₄-induced HCC microenvironment(C) (n=6).

(D) Tumor volume and tumor weight of subcutaneous Hepa1-6 tumors in C57BL/6 mice with NK cells depletion (treated with anti-NK1.1 neutralization antibody, n=8-11).

All data are presented as the mean \pm SEM from three independent experiments, and analyzed by unpaired Student's t test or two-way ANOVA with n.s., $p > 0.05$; *, $p < 0.05$; **, $p < 0.01$; ***, $p < 0.001$.

2. One of the main conclusions of this manuscript is that EE activates peripheral

SNS and β -AR signaling both in tumor cells and tumor infiltrating immune cells, leading to silencing of CCL2 expression and activation of anti-tumor immunity. However, the strategies and methodologies used to properly conclude that this is the mechanism behind the protective effects of eustress on HCC tumor growth are not convincing. First of all, the strategy used to determine the expression of ARs in the tumor cells vs cells of the tumor microenvironment is not correct and tumor cell characterization is based on the CD45 negative expression, which is not entirely correct, as this classification might also include other cell types. Besides, one of their main conclusions regarding the EE-induced reduction of CCL2 levels is mainly based on the quantification of serum levels of this chemokine, which might not be necessarily a reflection of what happens in the liver but also a systemic effect. In this regard, the source of CCL2 could be indicative of the beneficial effects of EE in other organs or systems and not only the liver. Moreover, concerning this aspect CCL2 expression in the tumor tissue is only shown by immunohistochemistry in the DEN+CCL4 model (Figure 3), which is not even quantified and by qPCR in the CCL4-induced HCC and DEN+HFD HCC models, in which there are not significant differences (Suppl. Figure 3C, D). mRNA levels of CCL2 expression by qPCR in the liver of mice with DEN-CCL4-induced liver carcinogenesis is not even shown.

Response: Thanks for the comment. In the tumor, CD45 negative cells are mostly tumor cells, but we do agree with the reviewer that it cannot be ruled out that they also include other cells. To avoid this confusion, we have made special notes in the revised manuscript.

For the CCL2 detection, we determined the expression of CCL2 protein both in blood serum (Fig. 3A) and liver tumor tissues (Fig. 3B and supplementary Fig.4F) from DEN+CCL₄ induced tumor model or s.c. tumor-bearing mice. We proved that EE could reduced CCL2 expression in both serum and tumor tissues (Fig. 3A,B). These results were again verified in the qRT-PCR (Supporting Fig.7A and also seen in Supplementary Fig.4G) and western-blotting (Fig.4E) analysis of DEN+CCL₄-induced and s.c. Hepa1-6 tumor models. Moreover, the expression of CCL2 in the spleen and bone marrow was also detected by ELSA assay from DEN+CCL₄-induced tumor mice under SE or EE conditions. The data showed that no change in CCL2 expression in the spleen, while EE significantly reduced CCL2 expression in the bone marrow (Supporting Fig.7B). We further proved that EE reduced the mRNA expression of CCL2 in the tumor tumor infiltrated TAMs and GMDSC from hepa1-6 tumor model (Fig.5H,I). Importantly, we verified that CCL2 is required for the EE induced anti-tumor immunity in the DEN+CCL₄-induced HCC model (Fig. 3E) and Hepa1-6 s.c tumor models (Fig. 3F). Blockade of CCL2 abrogated the EE induced inhibitory effect on both models (Supporting Fig.7C and also seen in Fig.3E,F).

As suggested, we made a careful statistical analysis on the mRNA expression of CCL2 and other chemokines in the tumor tissues from CCL₄-induced HCC model (Supporting Fig.7D; also seen in Supplementary Fig.4C) and DEN+HFD induced

HCC models (Supporting Fig.7E; also seen in Supplementary Fig.4D).
 These results have been added to the revised manuscript.

Supporting Fig.7 CCL2 is required for the EE induced anti-tumor immunity

(A) CCL2 mRNA expression was quantified by qRT-PCR in DEN+CCl₄ induced tumors or subcutaneous Hepa1-6 tumors under SE or EE feeding condition (n=5).
 (B) ELISA analysis of CCL2 in the spleen and bone marrow from DEN+CCl₄-treated mice which were fed under SE or EE conductions (n=4-10).
 (C) Total tumor number and diameter ≥ 3 mm tumor number in the liver of DEN+CCl₄-treated mice with CCL2 neutralized antibody under SE or EE feeding conditions. (n=5-10)
 (D, E) qPCR analysis of cytokines and chemokines mRNA expression in tumors tissues of CCl₄-induced (D) and DEN+HFD-induced (E) HCC models (n=5).
 All data are presented as the mean \pm SEM, and analyzed by unpaired Student's t test or two-way ANOVA with n.s., p>0.05; *, p<0.05; **, p<0.01; ***, p<0.001.

3. The author's primary method of flow cytometry is very poorly described.

Response: We apologise for this. We have redescribed the flow cytometry method as the following:

Flow Cytometry: Fresh mouse tumor tissues were harvested, minced, and digested into single cell with mouse tumor dissociation kits (Miltenyi Biotech) according to the manufacturer's instructions. First, the single-cell suspensions were centrifuged and suspended in stain buffer (BD Pharmingen) after removal of red blood cells, and then incubated with the anti-mouse CD16/32 antibody (BD Pharmingen) for 15 min to prevent non-specific binding. Second, cells were incubated with Fixable viability stain 510 (BD Pharmingen) to exclude the dead cells. After this step, cells were stained with all relevant antibodies for 1 h at room temperature away from the light. Then cells were washed twice with PBS and re-suspended in 200 μ L stain buffer. Last,

single-cell suspensions were analyzed by BD FACS Aria II . According to isotype and fluorescence-minus-one (FMO), gating strategies were as follows: CD8⁺ T cells (Live⁺CD45⁺CD3e⁺CD8⁺), CD4⁺ T cells (Live⁺CD45⁺CD3e⁺CD4⁺), M1-TAMs (Live⁺CD45⁺Ly6G⁻CD11b⁺F4/80⁺CD206⁻), M2-TAMs (Live⁺CD45⁺Ly6G⁻CD11b⁺F4/80⁺CD206⁺), G-MDSCs (Live⁺CD45⁺CD11b⁺Ly6G⁺), M-MDSCs (Live⁺CD45⁺CD11b⁺Ly6C⁺), and NK cells (Live⁺CD45⁺CD3e⁻NK1.1⁺). The data were analyzed with FlowJo software, and gating strategy was shown in Supplementary Fig.11, characterizing the immune cell infiltrates in tumor tissue. The FACs antibody used in the study was shown in the **Supplementary table 2**. **The flow cytometry method was fully described in the revised manuscript.**

4. Although authors develop different HCC mouse models which are based both on carcinogen-induced and on transplantable syngeneic liver tumors, the potential mechanism of sympathetic modulation of CCL2 expression via β -ARs should be fully addressed on the setting of carcinogen-induced liver cancer models, as the effects of the tumor microenvironment in this condition is not properly reproduced in subcutaneous syngeneic mouse models. In this regard, there are many aspects of the mechanism that are only partly addressed in the DEN+CCl₄ model. Once again, in one of the most important sections trying to unravel the mechanism of the sympathetic modulation of CCL2 expression via β -ARs, in which β -AR blockade is performed in DEN+CCl₄ model to analyze CCL2 levels, the protein levels of this chemokine are shown in the serum and by immunofluorescence and immunohistochemistry (Figure 5). The quality of the immunofluorescence is very poor and immunohistochemistry images are not quantified (Figure 5 C, D).

Response: Thanks for the valuable suggestions. We have added extra data in DEN+CCl₄ model to the manuscript to support our hypotheses and conclusions. Our latest results showed that, in DEN+CCl₄-induced HCC mouse model, EE reduces tumor growth dependent on CD8⁺ T cells, and depletion of CD8⁺ T cells abolished the EE-mediated tumor protective effect (**Fig.2I**). We also verified that CCL2 is required for the EE induced anti-tumor immunity in DEN+CCl₄-induced HCC model. Blockade of CCL2/CCR2 signaling with CCR2 KO mice (**Fig.3D**) or administration of CCL2 neutralized antibody (**Fig.3E**) abrogated the inhibitory effect of EE. We showed the upregulation of β -AR signaling in the DEN+CCl₄ induced tumor from EE mice (**Fig.4C-E**), and blockade of β -AR signaling abolished the antitumor protective effect of EE (**Fig.4F**). Moreover, our latest data further proved that blockade of β -AR signaling abolished the effect of EE-mediated overcome anti-PD-L1 resistance in DEN+CCl₄ (**Supplementary Fig.7D,E**). These results together with that from the s.c. tumor bearing mice could be better to support our hypotheses and conclusions.

For all immunohistochemistry results, we have added quantitative results using ImageJ (Please see detailed changes in Figs. 3-5 and related supplementary figures). And we have updated the manuscript with high-quality immunofluorescence images (**Fig.5A**). We added qRT-PCR analysis of the expression of CCL2 mRNA and ELISA analysis of the level of CCL2 in the tumor tissue extracts. All these data

showed that blockade of β -ARs signaling dramatically abrogated the EE-induced CCL2 reduction in the blood (**Fig.5D**) and tumor tissue (**Supporting Fig.8A, B, and also seen in Fig.5B,E,F**) in the DEN+CCl₄-induced HCC model.

These results have been added to the revised manuscript.

Supporting Fig.8 (Also seen in Fig.5E, F) Blockade of SNS/ β -ARs signaling abolishes EE-induced CCL2 reduction in tumor cells and immune cells

(A) ELISA analysis of CCL2 in the tumor extract from DEN+CCl₄-induced tumor-bearing mice under SE or EE feeding condition with or without β -AR blockade (n=4-6).

(B) mRNA expression of CCL2 in the liver from DEN+CCl₄-induced tumor-bearing mice under SE or EE feeding condition with or without β -AR blockade (n=5).

5. Some of the statistical comparisons between the different groups are not clear enough to this reviewer. Some of the examples include Figure 2F, Figure 3H, Figure 4E,F,G, Figure 6H,I Suppl Figure 2C and Suppl Figure 3G. Authors should explicitly indicate which are the groups compared in each graph.

Response: Thanks for the valuable suggestions. We have relabeled them to show the comparisons in a more precise way. Please refer to the revised figures for the corresponding modifications.

6. One of the main results in this study is that EE housing significantly reduces the DEN/CCl₄-induced CCL2 levels. Although this is only verified in terms of serum levels and it should be confirmed at the hepatic level, being CCL2 a chemokine that is key for monocyte-derived macrophage recruitment to the liver, it would be interesting to measure the total counts of CCR2-expressing macrophages in the liver after EE in the DEN/CCl₄-induced HCC model compared to mice that have been housed in an standard environment.

Response: We have included data for CCL2 on both protein and mRNA levels. These data showed that EE reduced the expression of CCL2 in DEN+CCl₄ induced tumors (**Fig.3E, Fig.4E, Fig.5A,B and Supplementary Fig.4G**). Here we would like to emphasize that G-MDSCs and TAMs also highly express CCL2, and EE also reduces the CCL2 expression in G-MDSCs and TAMs (**Fig.5H, I**).

As suggested, we also check the CCR2 expression in tumor infiltrated macrophage

in mice under SE or EE housing. In general, DEN+CCl₄ induced liver tumor tissue had a higher proportion of CCR2⁺ macrophages infiltration than normal liver from healthy mice (**Supporting Fig.9A,B**). We also found that EE reduced the level of CCR2 on the macrophage membrane surface in both wild-type and tumor-bearing mice (**Supporting Fig.9A,B**).

Supporting Fig.9 CCR2 expression in tumor infiltrated macrophage in mice under SE or EE housing.

(A, B) The proportion of CCR2⁺ cells in the gate of macrophages (CD11b⁺ F4/80⁺) were analyzed by flow cytometry from the liver of normal mice or liver tumor from DEN/CCl₄-induced HCC bearing mice (n=4) under SE or EE feeding conditions. All data are presented as the mean ± SEM from three independent experiments, and analyzed by unpaired Student's t test or two-way ANOVA with n.s., p>0.05; *, p<0.05; **, p<0.01.

7. The Discussion of the manuscript should highlight the relevance of this study, contextualizing their work according to the recent literature in this field, instead of enumerating or summarizing the results again.

Response: Thanks for the suggestions. We have carefully revised the discussion session as suggested.

Minor comments

- Regarding the HCC mouse models used in this manuscript chronic CCL4 administration is not a proper model of liver carcinogenesis.

Response: We have used different models of liver carcinogenesis in our manuscript, although the molecular mechanisms were primarily discussed in the DEN+CCl₄-induced model and the subcutaneous tumor model. Consistent with previous reported [16], chronic CCl₄ administration did induce a few liver cancer lesions in our mouse model.

- All the IHC images should include scale bars and should be quantified.

Response: We have updated with the high-quality IHC pictures. Scale bars have been added to all images, and all images have been quantified using Image J.

- In the experiments carried out with syngeneic transplantable tumors, besides showing the progression of tumor growth by measuring the tumor volume, authors should also include the final tumor weight as shown in Figure 1 (I, L), Figure 3 (F,H) and Suppl Figure 4(G,H).

Response: All the final tumor weights have been added in the manuscript.

- Fig 2D and 2E are not mentioned in the main text.

Response: We are sorry for the misleading information. We've put them in the right place in the article.

- In Figure 3B authors should quantify the expression levels of CCL2 by Western Blotting and additionally show the mRNA levels of this chemokine by qPCR. Similarly, regarding Suppl. Figure 3F, the results of the CCL2 protein levels shown by immunoblotting should also be confirmed by qPCR.

Response: We have included data for CCL2 on both protein and mRNA levels. These data showed that EE reduced the release and expression of CCL2 in DEN+CCl₄ induced tumors and s.c. tumors (**Supporting Fig.10; also seen in Fig.4E and Supplementary Fig.4G**).

These results have been added to the revised manuscript.

- In Figure 4C, the protein levels of β -ARs should be assessed by western blotting.

Response: We have added western-blotting analysis data and found that EE increased the expression of β -ARs in DEN+CCl₄ induced tumors and s.c. Hepa1-6 tumors (**Supporting Fig.10; also seen in Fig.4E**).

Supporting Fig.10 The expression of β 1-AR, β 2-AR, β 3-AR, and CCL2 in tumor tissue. Western blot assay for detecting the expression of β 1-AR, β 2-AR, β 3-AR, and CCL2 in tumor tissue lysates from DEN+CCl₄-induced HCC model and subcutaneous Hepa1-6 tumors (n=4)

from mice with SE or EE housing.

- In Figure 5A, authors claim “the extensive expression of β 1-AR and β 3-AR in both tumor cells and immune cells, and β 2-AR commonly expressed on immune cells”.

However, in Figure 5A authors do not include any tumor specific marker in the immunofluorescence assay and therefore, they cannot affirm this.

Response: We apologise for the misrepresentation. The CD45⁺ cells are mostly tumor cells in the tumor tissue, which agrees with the published study[17]. However, we cannot rule out the presence of other cell types. Thus, to avoid the confusion, we have made special notes in the manuscript.

- In Figure 6E, authors should confirm the results taking out the possible outlier in the EE+aPD-L1 group.

Response: The outlier in Fig. 6E is likely due to the interindividual variability in mice. The significance was unchanged even we removed the outlier and conducted the statistical test again.

- The manuscript should be thoroughly revised to correct typos and grammar mistakes.

Response: Thank you for your suggestion. We have revised the manuscript carefully to avoid any typos or grammar mistakes.

Reviewer #3 (Remarks to the Author): with expertise in neuro-immunology and cancer

In this manuscript, the authors demonstrate that EE inhibits the growth of carcinogen-induced liver neoplasias and transplantable syngeneic liver tumors. They show that EE activated peripheral SNS and β -ARs signaling in tumor cells and tumor infiltrated myeloid cells, leading to silencing of CCL2 expression and activation of anti-tumor immunity.

Overall, this is an important paper but at this point, the manuscript contains some overstatements, unclear concepts, unclear data presentation, editing and statistics issues, and, in general, it feels more like a collection of stories.

Response: We thank the reviewers for all the suggestions which are valuable for us to improve the manuscript. According to current research literatures, a paradoxical relationship is found between β -ARs activity and tumor control in the context of eustress or distress models. It seems that β -ARs activation contributed to the cancer-promoting effect of distress, whereas required in tumor-protective effect of eustress models. In this study, we took more effect to demonstrate a U-shape relationship between NE/EPI concentration (as β -ARs agonists) and tumor growth. That is, moderate activation of β -ARs by lower dose of NE/EPI could inhibited tumor growth, while overactivation of β -ARs by higher dose of NE/EPI promoted tumor growth in mice. We proved that, compared to distress, eustress such as EE might mildly modulate the β -ARs activation to boost an antitumor immunity and overcomes anti-PD-L1 resistance.

After a comprehensive and thorough revision, we added more data in the manuscript to support our hypotheses and conclusions. we hope the revised manuscript could be acceptable for you.

• The authors use M1 and M2 definition that has been challenged in recent years and maybe better to use the specific, functional, cell characterization. Also, the gating for myeloid cells is not clear. It appears as if Ly6G was used twice in the gating (Fig s9)

Response: We have recognised that the latest literatures used the new definition of M1/ M2-like macrophages and we have followed the latest deterministic criteria and have reflected this in the manuscript accordingly. We apologise for the incorrect display in **Supplementary Fig.11**, and now we have revised it.

According to isotype and fluorescence-minus-one (FMO), gating strategies used were as follows: CD8⁺ T cells (Live⁺CD45⁺CD3e⁺CD8⁺), CD4⁺ T cells (Live⁺CD45⁺CD3e⁺CD4⁺), (Live⁺CD45⁺Ly6G⁻CD11b⁺F4/80⁺CD206⁻), (Live⁺CD45⁺Ly6G⁻CD11b⁺F4/80⁺CD206⁺), (Live⁺CD45⁺CD11b⁺Ly6G⁺), (Live⁺CD45⁺CD11b⁺Ly6G⁺), (Live⁺CD45⁺CD11b⁺Ly6G⁺), and NK cells (Live⁺CD45⁺CD3e⁻NK1.1⁺). The data were analyzed with FlowJo software, and

gating strategies were shown in **Supplementary Fig.11**, characterizing the immune cell infiltration in the tumor tissue. The FACs antibodies used in the study were shown in the **Supplementary table 2**.

• **What is the difference between graphs 2B and 3G, 4J in terms of the effects on M1 (the graphs don't seem to demonstrate the same effect)?**

Response: Thanks for the comments. We repeated these experiments and analyzed the proportion of different tumor infiltrated immune cells from Hepa1-6 tumor bearing mice under SE or EE housing. The results are shown with high stability and consistency. In general, EE housing could increase the proportion of M1-TAMs but decrease that of M2-TAM in the tumor tissues (Supporting Fig.11A-C; also seen in Fig.2B, Fig.3G and Fig.4L).

These results have been updated in the revised manuscript, seen detail in Fig.2B, Fig.3G and Fig.4L

Supporting Fig.11 The proportion of different tumor infiltrated immune cells from Hepa1-6 tumor bearing mice under SE or EE housing

(A) Different infiltrated immune cells within tumor microenvironment were analyzed by flow cytometry from mice tumors in Hepa1-6 model under SE or EE housing (n=5 for each group).

(B) subcutaneous Hepa1-6 tumors in C57BL/6 mice treated with anti-CCL2 neutralized antibody 18 days after tumor implantation under SE or EE housing. Infiltrated immune cells in tumor microenvironment were analyzed by flow cytometry from Hepa1-6 tumors (n=8).

(C) Subcutaneous Hepa1-6 tumors bearing mice were injected s.c. with 6 OHDA (25 mg/kg, n=11-14). Percent of immune cells in tumors were detected by flow cytometry.

• **Please show the data for the NK cells. Especially since in previous publications NK was shown to play an important role.**

Response: Thanks for the comment. In this study, we found that CD8⁺ T cells depletion abolished the tumor protective effect of EE in Hepa1-6 (**Fig.2F**), H22 (**Fig.2G**) and LPC-H12 (**Fig.2H**) tumor models, and similar results showed in DEN+CCl₄ -induced HCC mouse model (**Supporting Fig.12A** and also seen in **Fig.2I**). Flow cytometry analysis showed NK cells had no change in SE and EE groups (Supporting Fig. 12B, C). Mice were treated with anti-NK1.1 depletion

antibodies in conjunction with SE or EE housing, while EE still controlled tumor growth after NK cell depletion (Supporting Fig.12D and also seen in Supplementary Fig.2F). Studies have reported that given NK cells alone could not change tumors effectively in mice HCC models[15].

Taken together, NK cells certainly play important roles in the immune response, however, in our study design, these cells may not be the main contributors to the tumor growth.

Supporting Fig.12 Environmental eustress reshapes tumor microenvironment and reduces tumor growth dependent on CD8⁺ T cells, but not NK cells.

(A) Total tumor number and diameter ≥ 3 mm tumor number in the liver of DEN+CCl₄-treated mice with CD8⁺T cells depletion under SE or EE feeding conditions. (n=5-10)

(B-C) The proportion of NK cells were analyzed by flow cytometry in subcutaneous Hepa1-6 tumor microenvironment(B and C) and DEN+CCl₄-induced HCC microenvironment(C) (n=6).

(D) Tumor volume and tumor weight of subcutaneous Hepa1-6 tumors in C57BL/6 mice with NK cells depletion (treated with anti-NK1.1 neutralization antibody, n=8-11).

All data are presented as the mean \pm SEM from three independent experiments, and analyzed by unpaired Student's t test or two-way ANOVA with n.s., $p > 0.05$; *, $p < 0.05$; **, $p < 0.01$; ***, $p < 0.001$.

• Please add a magnified image of the CCL2 staining (3B). In general, most of IH staining should be quantified.

Response: We have updated the images to show the (200 times magnification, 200x) and enlarged (400 times magnification, 400X) images (Supporting Fig.13A), and have performed quantitative analysis on the (200X) images (Supporting Fig.13B). As

suggested, In addition, we have quantified the staining for all immunohistochemistry results in the revised manuscript using Image J.

Supporting Fig.13 The CCL2 expression on liver tumor tissues from DEN/CCl₄-induced HCC model. (A) Representative image of immunostaining of CCL2 on liver tumor tissues from DEN/CCl₄-induced HCC model. Original magnification 20 x 10, Scale bar, 100 μ m; Original magnification 40 x 10, Scale bar, 100 μ m . (B)The relative CCL2 expression in 20 x 10 magnification was quantified by Image J analysis (n=10).

• **Another method of CCL2 quantification in the tumor will be useful (even mRNA)**

Response: We have added data on the level of protein and mRNA of CCL2, and these data showed that EE reduced the release and expression of CCL2 in DEN+CCl₄ induced tumors and s.c. Hepa1-6 tumors (**Fig.4E and Supplementary Fig.4G**).

We added qRT-PCR analysis of the expression of CCL2 mRNA and ELISA analysis of the level of CCL2 in the tumor tissue extracts, these data showed that blockade of β -ARs signaling dramatically abrogated the EE-induced CCL2 reduction in the blood (**Fig.5D**) and tumor tissue (**Fig.5B, E, F**) in the DEN+CCl₄-induced HCC model.

• **Fig 4: The authors argue that the NE effect is local but this is not shown directly. It is possible for example that the effect is mediated via changes in the bone marrow innervation.**

Response: Thanks for the comment. To determine whether NE/EPI effect is local or not, the concentrations of NE/EPI and in the blood serum, bone marrow, spleen and the liver/liver tumor tissue were measured from the wild-type or DEN+CCl₄ tumor-bearing mice under SE or EE housing. The results showed EE housing could increase the level of serum NE and EPI in both naïve mice (**Supporting Fig.13A,C and also seen in Fig.4A**) and tumor bearing mice (**Supporting Fig.13B,D and also seen in Fig.4B**). Moreover, the NE and EPI in the tumor tissue were significantly increased in the DEN+CCl₄ models (**Supporting Fig.13B,D and also seen in Fig.4B**). Interestingly, NE in the bone marrow and EPI in the spleen were selectively increased in the DEN+CCl₄ models but not in naïve mice (**Supporting Fig.13A-D**). Given that β 1-AR, β 2-AR, β 3-AR were expressed in tumor tissue (**Fig. 4C, E**) and tumor cells as well as tumor infiltrated TAMs and G-MDSCs (**Supplementary Fig.5F-H**), the expression of which further enhanced under the EE housing, we speculated that the

tumor microenvironment might be the major place for action of NE and EPI. Moreover, previous study had shown the effect of NE on modulate the function of G-MDSCs in bone marrow[18], and we also proved that EE could the upregulate the level of NE in the bone marrow, indicating that bone marrow might be also another place for the G-MDSCs function modulation by NE.

We have revised the manuscript and provided further evidences to support our hypothesis.

Supporting Fig.13 The level of NE and EPI in the blood serum, bone marrow, spleen and liver/liver tumor tissue from normal mice or DEN+CCl₄-treated mice

(A, B) ELISA analysis of level of EPI in the blood serum, bone marrow, spleen and liver/liver tumor tissue from normal mice (A) or DEN+CCl₄-treated mice (B) which were fed under SE or EE condictions (n=6-11).

(C, D) ELISA analysis of level of NE in the blood serum, bone marrow, spleen and liver/liver tumor tissue from normal mice (C) or DEN+CCl₄-treated mice (D) which were fed under SE or EE condictions (n=6-11).

• 4H, why didn't the author made the distinction between M1 and M2

Response: Thanks for the comment. We have repeated the experiments and the result were consistent with the previous findings. Flow cytometry analysis also indicated a significant increase of CD8⁺ T cells and a dramatic decrease of M-MDSCs and G-MDSCs and M2-TAMs in EE mice compared to those in SE mice. Blockade of β-ARs signaling abrogated the EE-mediated increase of CD8⁺ T cells and decrease of G-MDSCs and M2-TAMs in the tumor microenvironment of either DEN/CCl₄ tumor model (Supporting Fig.14A and also seen in revised Fig.4J).

We have revised the manuscript and the updated data were shown in Fig.4J.

Supporting Fig.14 Percent of immune cells in tumors were detected by flow cytometry from DEN+CCl₄-tumor bearing mice. Mice were fed under SE or EE conditions with or without β-ARs blockade treatment (β-block: SR59230A+propranolol, n=6).

• **5A- the staining is not clear at all. What is the condition (EE)? What is the comparison?**

Response: We have repeated the IF staining with a higher resolution and the conditions have been clearly labeled in the new images. We analyzed the in situ immunostaining of CCL2, β1-AR, β2-AR, β3-AR and CD45 in tumor tissue from DEN+CCl₄ mouse models under SE or EE housing (**Supporting Fig.15A and also seen in Fig.5A and Supplementary Fig.6A**). The results revealed the extensive expression of β1-AR, β2-AR and β3-AR in both CD45⁻ cells and CD45⁺ immune cells (**Supporting Fig.15A and also seen in Fig.5A**). Moreover, CCL2, co-localized with β-ARs, were also expressed in both CD45⁻ cells and CD45⁺ immune cells (**Supporting Fig.15A and also seen in Fig.5A**), and it was consistent with our finding that both tumor-derived and immune cells-derived CCL2 signaling were required for EE-induced anti-tumor immunity.

We have revised the manuscript and the updated data were shown in Fig.5A and Supplementary Fig.6A.

Supporting Fig.15 Immunofluorescence staining of β1-AR, β2-AR, β3-AR, CCL2, CD45, and DAPI in liver tumor tissues from DEN+CCl₄-induced tumor-bearing mice under SE or EE feeding condition.

• 5E- the definition of tumor cells as CD45-. It does not make sense. These cells can be fibroblasts, for example.

Response: Thanks for the comment. In the tumor, CD45 Negative cells are mostly tumor cells, but we do agree with the reviewer that it cannot be ruled out that they also include other cells. To avoid this confusion, we have made special notes in the revised manuscript.

• Do the levels of CCR2 change?

Response: As suggested, we analyzed the CCR2 expression on the tumor tissue and on the tumor infiltrated immune cells from DEN+CCl₄-induced tumor-bearing mice under SE or EE feeding condition. The CCR2 mRNA expression in the tumors tissue in EE mice showed a significant decrease of compared to those in SE mice (**Supporting Fig.16A**). Next, we analyzed the proportion of CCR2 positive-TAM or -G-MDSCs in the total TAMs or G-MDSCs in the tumor microenvironment. Flow cytometry analysis showed EE decreased the expression of CCR2 in TAMs, but not the G-MDSCs (**Supporting Fig.4B,C**). Previously, we had proved that EE could significantly reduce the infiltration of total TAMs and MDSCs (**Fig. 2A**). Altogether, these results indicated that the decrease expression of CCR2 in the tumor tissue might result from the reduced total CCR2⁺ tumor infiltrated TAMs and G-MDSCs.

Supporting Fig.16 CCR2 expression in tumor tissue and infiltrated macrophage in mice under SE or EE housing.

(A) The mRNA expression of CCR2 in tumor tissues from DEN/CCl₄-induced HCC bearing mice under SE or EE housing (n=4-5). (B,C) The proportion of CCR2⁺ cells in the gate of macrophages (CD11b⁺ F4/80⁺) and G-MDSC (CD11b⁺ F480⁻ Ly6G⁺ Ly6C^{/low}) were

analyzed by flow cytometry from liver tumor from DEN/CCl4-induced HCC bearing mice (n=4) under SE or EE feeding conditions.

All data are presented as the mean \pm SEM, and analyzed by unpaired Student's t test or two-way ANOVA with n.s., $p > 0.05$; *, $p < 0.05$; **, $p < 0.01$.

• **What happens to CCL2 in CD8 cells in the EE?**

Response: Flow cytometry analysis revealed that there are three clusters of CCL2 in the DEN+CCl4-induced tumors, namely CCL2^{Hi}, CCL2^{Md} and CCL2^{Lo}. CCL2^{Hi} and Ccl2^{Lo} are mostly CD45⁺ immune cells, while CCL2^{Md} are mainly non-immune cells. In addition, CCL2^{Hi} was mainly CD11b⁺ marrow derived cells, mainly including G-MDSC and M-MDSC. 77.7% of CCL2^{Lo} cells were CD3e⁺ cells. Hence, majority of CD8⁺ T cells have low or no CCL2 expression (**Supporting Fig.17A**). We calculated the proportions of CCL2^{Hi} and CCL2^{Lo} in total cells of mice under SE and EE feeding conditions, and the results showed that EE reduced the proportion of CCL2^{Hi}, but did not change the proportion of CCL2^{Lo} (**Supporting Fig.17B**). EE significantly increased the proportion of CD8⁺ T cells (**Supporting Fig.17C**).

Supporting Fig.17 The CCL2 expression in different cells of tumor microenvironment from DEN+CCl₄-induced tumor-bearing mice under SE or EE feeding condition.

(A) Representative flow cytometric analyses the gate of CCL2 intracellular staining in suspension cells from DEN+CCl₄ induced tumor tissue.

(B) The proportion of CCL2^{Hi} and CCL2^{Lo} cells in the gate of total live cells were analyzed by flow cytometry in DEN+CCl₄-induced tumor microenvironment in mice under SE or EE feeding condition (n=4).

(C) The proportion of CD8 T cells (CCL2^{Lo} cells) were analyzed by flow cytometry in DEN+CCl₄-induced tumor microenvironment in mice under SE or EE feeding condition (n=4).

• **Supp. Fig 2. – it is not clear what the images indicates**

Response: We have repeated the IHC staining with a higher resolution and the conditions have been clearly labeled in the new images. Immunohistochemistry staining of CD4, CD8, F4/80 and Ly6G on DEN+CCl₄ tumors further confirmed the reshaping of the tumor microenvironment (**Supporting Fig.18A, B and also seen in Supplementary Fig.2A, B**). In EE mice, showed a significant increase of CD8⁺ T cells and dramatic decrease of G-MDSCs and TAMs compared to those in SE mice (**Fig.2A, D**). The numbers of other immune cells, such as CD4⁺ T cell exhibited no appreciable changes.

Supporting Fig.17 Immunostaining of tumor infiltrated immune cells

(A) Immunostaining of CD4, CD8, F4/80, and Ly6G on liver tumor tissues from DEN+CCl₄-induced HCC model under SE or EE feeding conditions. Original magnification 20 x 10, Scale bar, 50 μm.

(B) Positive areas of CD4, CD8, F4/80, and Ly6G on liver tumors tissues area were quantified by Image J analysis in DEN+CCl₄-induced HCC mouse model (n=6).

• **Many supplementary graphs are not clear (for example, Fig s3G). 3C, D – is missing statistics.**

Response: We are sorry for the unclear information. As suggested, we have gone through a comprehensive and thorough revision, and hope the revised manuscript could be acceptable for you.

• **In general, it may be useful to perform a statistical review of the paper**

Response: Thanks for the useful suggestion. We have carefully checked all the statistical analysis of this manuscript and re-labeled the groups with statistical

significance.

- **The authors refer to norepinephrine and epinephrine as hormones, but they are not classically considered hormones, so this point requires clarification.**

Response: Thanks for the comment. It is true that norepinephrine and epinephrine represent two neurotransmitters more than classically hormones. They usually function as stress hormones, and contribute to the catecholamine family of chemicals. They affect various areas of the body and activate the central nervous system like hormones under eustress or distress. As suggested, we specialized the norepinephrine and epinephrine as “Stress Hormones” in the Fig.6L.

- **The connection of PDL-1 should be explained in more detail.**

Response: Thanks for the comment. Recent studies have shown the sympathetic and parasympathetic nerves in tumor tissues were correlated with the expression of PD-1/PD-L1[11]. As previously discussed in our manuscript, EE relieved the tumor immunosuppression and boosted the antitumor immunity via SNS/ β -ARs/CCL2, which drove a hypothesis that EE is likely to help overcome PD-L1 resistance. Indeed, we demonstrated that EE housing increases the efficacy of anti-PD-1 via activating β -AR signaling. We further tested the effect of β -AR agonists on therapeutic efficacy of anti-PD-1. We showed a U-shape relationship between NE/EPI concentration (as β -ARs agonists) and tumor control. Our data showed that low dose of NE or EPI (2 mg/kg) could mimic the anti-tumor and immunomodulatory effects of EE (Supporting Fig.18A,B; also seen in Supplementary Fig. 6I,J). Of note, mice in the NE+ α PD-L1 or EPI+ α PD-L1 groups showed a robust tumor control indicated by the smallest tumor size, suggesting that β -ARs agonists could improve the efficacy of anti-PD-1 (Supporting Fig.18 A,B). Blockade of β -AR signaling abolished the effect of EE-mediated overcome anti-PD-L1 resistance in DEN+CCl₄ (Supporting Fig.18C,D) and LPC-H12 tumor models (Supporting Fig.18E,F), but had no obvious effect on the therapeutic effect of anti-PD-L1 in SE mice (Supporting Fig.18E,F). Altogether, these results confirmed that EE overcomes anti-PD-L1 resistance via modulating β -ARs signalings.

These results have been added to the revised manuscript. Please see detailed changes in Supplementary Fig. 7B-G.

Supporting Fig. 18 (Also seen in Supplementary Fig. 7) Environmental eustress enhances tumor β -AR signaling to augment the therapeutic of PD-L1/PD-1 blockade.

(A) Scheme of experimental procedure for subcutaneous LPC-H12 tumor model with NE or EPI treatment (2mg/kg) combined with anti-PD-L1 immunotherapy.

(B) Tumor volume and tumor weight of C57BL/6 mice bearing subcutaneous LPC-H12 tumors with NE or EPI treatment plus anti-PD-L1 immunotherapy (n=6).

(C) Scheme of experimental procedure for DEN+CCl₄-induced tumor model with or without anti-PD-L1 immunotherapy, or/and β -ARs blockade treatment (β -block: SR59230A+propranolol).

(D) Total tumor numbers (left) and numbers of tumor with diameter $\phi \geq 3$ mm (right) on livers from DEN+CCl₄-induced tumor-bearing mice with or without anti-PD-L1 immunotherapy, or/and β -ARs blockade treatment (n=4-10).

(E) Scheme of experimental procedure for subcutaneous LPC-H12 tumor model treated with or without anti-PD-L1 immunotherapy, or/and β -ARs blockade.

(F) Tumor volume and tumor weight of C57BL/6 mice bearing subcutaneous LPC-H12 tumors treated with or without anti-PD-L1 immunotherapy, or/and β -ARs blockade (n=10-12).

All data are presented as the mean \pm SEM from three independent experiments, and analyzed by two-way ANOVA with n.s., $p > 0.05$, *, $p < 0.05$; **, $p < 0.01$; ***, $p < 0.001$.

- **Many studies show that beta-blockers are in fact protective against tumors, therefore the argument that it's the opposite, requires more careful discussion and integration with the existing literature.**

Response: The Sympathetic nervous system (SNS) is commonly associated with the stress response, including distress and eustress. In response to stress and activation of SNS, NE and EPI were upregulated and stimulated β -ARs locally and systematically, leading to both advantageous and harmful effects on organisms depending on the duration of the response and other unknown factors[1, 2]. According to current research literatures, a paradoxical relationship is found between β -ARs activity and tumor control in the context of eustress or distress models. It seems that β -ARs activation contributed to the cancer-promoting effect of distress[3-5], whereas required in tumor-protective effect of eustress models[6-8]. Previous study had shown voluntary running, one of the major components in EE, dramatically reduced the melanoma cell dissemination and lung metastasis in a Epinephrine dependent manner[9]. Blockade of β -ARs signaling blunts the exercise-induced tumor suppression[9]. These results indicated the activation of β -ARs might functionally differ on cancer biology in a context-dependent and non-linear manner.

In this study, **we found a U-shape relationship of NE/EPI concentration with CCL2 expression in human liver organoid, cultured tumor cells, TAMs or MDSCs, that is, lower concentration of NE/EPI help to reduce the CCL2 expression, which higher concentration subverted this effect (Supporting Fig.19A-E and also seen in Supplementary Fig.6E-I).** Studies on running inhibition of tumors in mice have shown that EPI injection daily with a low-dose of 0.5mg/kg or 2mg/kg for several days could inhibit s.c. tumor growth, mimicking the effect of Voluntary exercise[8]. While chronic treatment with β -agonist isoprenaline at a dose of 10mg/kg daily promoted tumor development and impaired the antitumor -immunity [10]. Our results showed that 0.5mg/kg and 2mg/kg NE/EPI inhibited s.c. tumor growth, but 6mg/kg EPI or 8mg/kg NE promoted tumor growth in mice (**Supporting Fig.19F, G and Supplementary Fig.5K, 7B**). These data suggest that specific functions of EPI/NE require specific doses.

Taken together, we hypothesized that it was the U-shape relationship of NE/EPI concentration that led to the differences in β -ARs and agonists/antagonists shown in studies, the mechanism of which would be an interesting study worth pursuing further.

Supporting Fig.19 The U-shape relationship between NE/EPI concentration, CCL2 expression, and tumor growth.

(A) Human hepatocytes/hepatic stellate cells organoids were treated with varied concentration of NE and EPI (ng/ml) in vitro for 24h, followed by washing and medium replacement. 48 h later, CCL2 mRNA expression was determined with qPCR assay with β -actin as an internal control (n=3).

(B, C) The mRNA expression of CCL2 was determined with qPCR assay in Hepa1-6 (B) and LPC-H12 (C) cells after a 24h-treatment with varied concentration of NE and EPI (50 or 500 ng/ml) in vitro (n=3).

(D) Bone marrow derived macrophages (BMDMs) were exposed to conditioned medium (CM) of Hepa1-6 tumor cells for 48h and subsequently treated with vehicle, NE and EPI (0.5, 5, 50, 500 ng/ml) in vitro for 24h. Cells were washed followed by medium replacement. 48h later, the mRNA expressions of CCL2 in BMDM were determined

(E) Bone marrow cells were isolated from normal C57BL/6 mice and cultured in the presence of recombinant murine granulocyte macrophage colony-stimulating factor (GM-CSF) for up to 7 days. Ly6G⁺ G-MDSCs were sorted out and subsequently treated with vehicle, NE and EPI (0.5, 5, 50, 500 ng/ml) in vitro. The mRNA expressions of CCL2 in G-MDSCs were determined after treatment (n=3).

(F, G) Tumor volume of subcutaneous LPC-H12 tumors in mice injected s.c. with different doses of NE or EPI every three days under SE feeding conditions (n=4-6).

• In addition, many studies refer to b2 receptor in the context of the tumor but here, the authors link their work mainly the b3 and b1 receptor. This gap should

be discussed.

Response: Thanks for the comments. In this study, our data showed that EE housing could robustly increase the expression of β 1-AR and β 3-AR and moderately upregulate the expression of β 2-AR in the tumor tissue from DEN+CCl₄ model (**Fig.4D, E**) and subcutaneous tumor model (**Fig.4E**). The expression of β -ARs was further separately determined in the CD45⁻ cells, TAMs and G-MDSCs in the tumor microenvironment. The results revealed that EE elevated the β 1-AR and β 3-AR mRNA expression level in CD45⁻ cells (mainly tumor cells, **Supplementary Fig.5F**) and TAMs (**Supplementary Fig.5G**) and G-MDSCs (**Supplementary Fig.5H**). β 2-AR mRNA seemed to be only upregulated in CD45⁻ cells and TAMs, but not G-MDSCs (**Supplementary Fig.5F-H**).

Moreover, we showed EE housing could increase the level of NE and EPI, the β -AR agonist, in the circulation and tumor tissues. Mice in EE fed the β -blocker (PROP+SR, blocking all β -ARs) lost the protection against tumor (**Fig.4F-H**) and abolished the EE-mediated CCL2 reduction (**Fig.5B-F**). Thus, enhanced β -AR signaling at the ligand and receptor levels mediates potent benefits of EE against tumor in vivo. Consistently, previous study also showed EE enhanced chronic activation of β -AR signaling at both ligand and receptors level to prevent microglia inflammation by amyloid- β and provided protection against features of Alzheimer's disease[19].

According to current research literatures, a paradoxical relationship is found between β -ARs activity and tumor control. It seems that β -ARs activation contributed to the cancer-promoting effect of distress[3-5], whereas required in tumor-protective effect of eustress models[6-8]. The possible mechanisms might include the distinct patterns of tissue distribution of β -ARs (e.c. β 1-AR, β 2-AR and β 3-AR) and signal through distinct biochemical pathways, which functionally differ on cancer biology in a context-dependent and non-linear manner.

We have discussed the distinct functions of β -ARs signalings in the revised manuscript.

• **The tumor models are not described in sufficient detail.**

Response: We have supplemented the description of the tumor model.

• **To what extent the effects is the EE and not just physical activity?**

Response: Environment Enrichment (EE) is an established environmental eustress model for giving mice more social interaction in a large activity space, running wheels for sports, and other toys for hiding and playing [20-22]. To investigate whether physical exercise could alone account for the effect of EE, we have removed of any other toys and have provided the mice a running wheel on which they can move freely. However, the results showed that mice with more physical activity did not delay the progression of Hepa1-6 tumor effectively (**Supporting Fig.20A**). Moreover, anti-tumor effect of voluntary running was also decreased compared to that of EE in the DEN+HFD-induced tumor model (**Supporting Fig.20B**). These

data suggested that EE is the main contributor to the anti-tumor effects and physical activities might partially contribute to the protective effect of EE [20, 23].

Supporting Fig.20 The anti-tumor effects of EE are not limited to running

(A) Tumor volume of subcutaneous Hepa1-6 (n=8-10) and (B) Total tumor number and diameter $\phi \geq 3$ mm tumor number in mice treated with DEN+HFD under SE, EE or voluntary running feeding conditions (n=8).

• **The authors indicate that EE attenuates growth, but it can also be reduced insemination.**

Response: We are very sorry that we may not have fully understood this comment. Maybe the reviewer wondered the effect of tumor dissemination and tumor metastasis.

Our study reveals that environmental eustress via EE stimulates antitumor immunity for better tumor control and immunotherapy. We found that EE activated peripheral SNS and β -ARs signaling in tumor cells and tumor infiltrated myeloid cells, leading to suppression of CCL2 expression and activation of anti-tumor immunity. We and others had reported that CCL2 is critical determinant for both tumor metastasis and immunosuppression in HCC[24], breast cancer[25], colon cancer[26, 27]. Previous study had shown voluntary running, one of the major components in EE, dramatically reduced the melanoma cell dissemination and lung metastasis in a Epinephrine dependent manner[9]. Blockade of β -ARs signaling blunts the exercise-induced tumor suppression[9]. These results indicate that EE might exert protection against tumor growth and metastasis via activating β -ARs/CCL2 signaling. Further studies are needed to demonstrate this function.

References

- [1] J.W. Eng, K.M. Kokolus, C.B. Reed, B.L. Hylander, W.W. Ma, E.A. Repasky, A nervous tumor microenvironment: the impact of adrenergic stress on cancer cells, immunosuppression, and immunotherapeutic response, *Cancer immunology, immunotherapy* : CII, 63 (2014) 1115-1128.
- [2] S.W. Cole, A.S. Nagaraja, S.K. Lutgendorf, P.A. Green, A.K. Sood, Sympathetic nervous system regulation of the tumour microenvironment, *Nature reviews. Cancer*, 15 (2015) 563-572.
- [3] M.J. Bucsek, G. Qiao, C.R. MacDonald, T. Giridharan, L. Evans, B. Niedzwecki, H. Liu, K.M. Kokolus, J.W. Eng, M.N. Messmer, K. Attwood, S.I. Abrams, B.L. Hylander, E.A. Repasky, beta-Adrenergic Signaling in Mice Housed at Standard Temperatures Suppresses an Effector Phenotype in CD8(+) T Cells and Undermines Checkpoint Inhibitor Therapy, *Cancer research*, 77 (2017) 5639-5651.
- [4] J.W. Eng, C.B. Reed, K.M. Kokolus, R. Pitoniak, A. Utley, M.J. Bucsek, W.W. Ma, E.A. Repasky, B.L. Hylander, Housing temperature-induced stress drives therapeutic resistance in murine tumour models through beta2-adrenergic receptor activation, *Nature communications*, 6 (2015) 6426.
- [5] H. Mohammadpour, M.J. Bucsek, B.L. Hylander, E.A. Repasky, Depression Stresses the Immune Response and Promotes Prostate Cancer Growth, *Clinical cancer research : an official journal of the American Association for Cancer Research*, 25 (2019) 2363-2365.
- [6] L. Cao, X. Liu, E.J. Lin, C. Wang, E.Y. Choi, V. Riban, B. Lin, M.J. During, Environmental and genetic activation of a brain-adipocyte BDNF/leptin axis causes cancer remission and inhibition, *Cell*, 142 (2010) 52-64.
- [7] Y. Song, Y. Gan, Q. Wang, Z. Meng, G. Li, Y. Shen, Y. Wu, P. Li, M. Yao, J. Gu, H. Tu, Enriching the Housing Environment for Mice Enhances Their NK Cell Antitumor Immunity via Sympathetic Nerve-Dependent Regulation of NKG2D and CCR5, *Cancer research*, 77 (2017) 1611-1622.
- [8] L. Pedersen, M. Idorn, G.H. Olofsson, B. Lauenborg, I. Nookaew, R.H. Hansen, H.H. Johannesen, J.C. Becker, K.S. Pedersen, C. Dethlefsen, J. Nielsen, J. Gehl, B.K. Pedersen, P. Thor Straten, P. Hojman, Voluntary Running Suppresses Tumor Growth through Epinephrine- and IL-6-Dependent NK Cell Mobilization and Redistribution, *Cell metabolism*, 23 (2016) 554-562.
- [9] L. Pedersen, M. Idorn, G.H. Olofsson, B. Lauenborg, I. Nookaew, R.H. Hansen, H.H. Johannesen, J.C. Becker, K.S. Pedersen, C. Dethlefsen, J. Nielsen, J. Gehl, B.K. Pedersen, P.T. Straten, P. Hojman, Voluntary Running Suppresses Tumor Growth through Epinephrine- and IL-6-Dependent NK Cell Mobilization and Redistribution, *Cell Metab*, 23 (2016) 554-562.
- [10] M.D. Nissen, E.K. Sloan, S.R. Mattarollo, beta-Adrenergic Signaling Impairs Antitumor CD8(+) T-cell Responses to B-cell Lymphoma Immunotherapy, *Cancer Immunol Res*, 6 (2018) 98-109.
- [11] A. Kamiya, Y. Hayama, S. Kato, A. Shimomura, T. Shimomura, K. Irie, R. Kaneko, Y. Yanagawa, K. Kobayashi, T. Ochiya, Genetic manipulation of autonomic nerve fiber innervation and activity and its effect on breast cancer progression, *Nature Neuroscience*, 22 (2019) 1289-1305.
- [12] D.I. Gabrilovich, S. Nagaraj, Myeloid-derived suppressor cells as regulators of the immune system, *Nature Reviews Immunology*, 9 (2009) 162-174.
- [13] D.G. DeNardo, B. Ruffell, Macrophages as regulators of tumour immunity and immunotherapy, *Nature Reviews Immunology*, 19 (2019) 369-382.
- [14] V. Kumar, L. Donthireddy, D. Marvel, T. Condamine, F. Wang, S. Lavilla-Alonso, A. Hashimoto, P. Vonteddu, R. Behera, M.A. Goins, C. Mulligan, B. Nam, N. Hockstein, F. Denstman, S. Shakamuri, D.W. Speicher, A.T. Weeraratna, T. Chao, R.H. Vonderheide, L.R. Languino, P. Ordentlich, Q. Liu, X. Xu, A. Lo, E. Pure, C. Zhang, A. Loboda, M.A. Sepulveda, L.A. Snyder, D.I. Gabrilovich, Cancer-Associated Fibroblasts Neutralize the Anti-tumor Effect of CSF1 Receptor Blockade by Inducing PMN-MDSC Infiltration of

Tumors, *Cancer Cell*, 32 (2017) 654-+.

- [15] M. Kim, S.-J. Lee, S. Shin, K.-S. Park, S.Y. Park, C.H. Lee, Novel natural killer cell-mediated cancer immunotherapeutic activity of anisomycin against hepatocellular carcinoma cells, *Scientific Reports*, 8 (2018) 10668.
- [16] X. Zhao, J. Fu, A. Xu, L. Yu, J. Zhu, R. Dai, B. Su, T. Luo, N. Li, W. Qin, B. Wang, J. Jiang, S. Li, Y. Chen, H. Wang, Gankyrin drives malignant transformation of chronic liver damage-mediated fibrosis via the Rac1/JNK pathway, *Cell Death & Disease*, 6 (2015) e1751-e1751.
- [17] B.I. Reinfeld, M.Z. Madden, M.M. Wolf, A. Chytil, J.E. Bader, A.R. Patterson, A. Sugiura, A.S. Cohen, A. Ali, B.T. Do, A. Muir, C.A. Lewis, R.A. Hongo, K.L. Young, R.E. Brown, V.M. Todd, T. Huffstater, A. Abraham, R.T. O'Neil, M.H. Wilson, F. Xin, M.N. Tantawy, W.D. Merryman, R.W. Johnson, C.S. Williams, E.F. Mason, F.M. Mason, K.E. Beckermann, M.G. Vander Heiden, H.C. Manning, J.C. Rathmell, W.K. Rathmell, Cell-programmed nutrient partitioning in the tumour microenvironment, *Nature*, 593 (2021) 282-288.
- [18] T.L. Ben-Shaan, M. Schiller, H. Azulay-Debby, B. Korin, N. Boshnak, T. Koren, M. Krot, J. Shakya, M.A. Rahat, F. Hakim, A. Rolls, Modulation of anti-tumor immunity by the brain's reward system, *Nat Commun*, 9 (2018) 2723.
- [19] H. Xu, M.M. Rajsombath, P. Weikop, D.J. Selkoe, Enriched environment enhances beta-adrenergic signaling to prevent microglia inflammation by amyloid-beta, *EMBO Mol Med*, 10 (2018).
- [20] L. Cao, X. Liu, E.-J.D. Lin, C. Wang, E.Y. Choi, V. Riban, B. Lin, M.J. During, Environmental and Genetic Activation of a Brain-Adipocyte BDNF/Leptin Axis Causes Cancer Remission and Inhibition, *Cell*, 142 (2010) 52-64.
- [21] Y. Song, Y. Gan, Q. Wang, Z. Meng, G. Li, Y. Shen, Y. Wu, P. Li, M. Yao, J. Gu, H. Tu, Enriching the Housing Environment for Mice Enhances Their NK Cell Antitumor Immunity via Sympathetic Nerve-Dependent Regulation of NKG2D and CCR5, *Cancer Research*, 77 (2017) 1611-1622.
- [22] A.M. Slater, L. Cao, A Protocol for Housing Mice in an Enriched Environment, *Jove-Journal of Visualized Experiments*, (2015).
- [23] G. Li, Y. Gan, Y. Fan, Y. Wu, H. Lin, Y. Song, X. Cai, X. Yu, W. Pan, M. Yao, J. Gu, H. Tu, Enriched Environment Inhibits Mouse Pancreatic Cancer Growth and Down-regulates the Expression of Mitochondria-related Genes in Cancer Cells, *Scientific Reports*, 5 (2015) 7856.
- [24] X.G. Li, W.B. Yao, Y. Yuan, P.Z. Chen, B. Li, J.Q. Li, R.A. Chu, H.Y. Song, D. Xie, X.Q. Jiang, H. Wang, Targeting of tumour-infiltrating macrophages via CCL2/CCR2 signalling as a therapeutic strategy against hepatocellular carcinoma, *Gut*, 66 (2017) 157-167.
- [25] B.Z. Qian, J.F. Li, H. Zhang, T. Kitamura, J.H. Zhang, L.R. Campion, E.A. Kaiser, L.A. Snyder, J.W. Pollard, CCL2 recruits inflammatory monocytes to facilitate breast-tumour metastasis, *Nature*, 475 (2011) 222-U129.
- [26] M.J. Wolf, A. Hoos, J. Bauer, S. Boettcher, M. Knust, A. Weber, N. Simonavicius, C. Schneider, M. Lang, M. Sturzl, R.S. Croner, A. Konrad, M.G. Manz, H. Moch, A. Aguzzi, G. van Loo, M. Pasparakis, M. Prinz, L. Borsig, M. Heikenwalder, Endothelial CCR2 Signaling Induced by Colon Carcinoma Cells Enables Extravasation via the JAK2-Stat5 and p38MAPK Pathway, *Cancer Cell*, 22 (2012) 91-105.
- [27] L. Zhao, S.Y. Lim, A.N. Gordon-Weeks, T.T. Tapmeier, J.H. Im, Y.H. Cao, J. Beech, D. Allen, S. Smart, R.J. Muschel, Recruitment of a Myeloid Cell Subset (CD11b/Gr1(mid)) via CCL2/CCR2 Promotes the Development of Colorectal Cancer Liver Metastasis, *Hepatology*, 57 (2013) 829-839.

Reviewers' Comments:

Reviewer #1:

Remarks to the Author:

The authors have addressed most of my concerns and have conducted extensive additional experimentation which has added greatly to the value of this study.

Reviewer #2:

Remarks to the Author:

Authors have adequately addressed the questions raised in the first revision as well as performed new experimental mouse models.

Minor Comments

- Western Blottings were not quantified in Figure 4E. In order to properly conclude immunoblottings need to be quantified.

- Authors have incorporated a new figure as Figure 3E. In this Figure the total tumor number and the number of tumors with more than 3mm diameter in the liver of DEN+CCl4 mice treated with anti-CCL2 were included. Please in order to be consistent with all the figures please include above the two graphs in panel E that this mice are DEN+CCl4. Same applies for Figure 2E. in line with this, in those panels in which tumor volume and tumor weight are included, include the mouse model above both graphs or in one of them but it needs to be consistent in all the figures. Some examples include Figure 2F, Figure 2G, Figure 2H, Figure 3F and Figure 3H.

Reviewer #3:

Remarks to the Author:

Overall the authors did a significant attempt to address my comments. I am still puzzled by the lack of effects on NK but the new data is supportive of this statement. It will be helpful to expand the discussion of this point.

For supp Fig. 15 it will be helpful to get an insert of a magnified image.

Response to Reviewers

We would like to thank you for all your constructive comments. They are all very helpful for us to further improve and strengthen our manuscript. As suggested, we have performed additional changes accordingly to address your concerns and have responded point-by-point to each of the comments as the following:

Reviewer #1 (Remarks to the Author):

The authors have addressed most of my concerns and have conducted extensive additional experimentation which has added greatly to the value of this study.

Response: Thanks for all your efforts throughout this review process.

Reviewer #2 (Remarks to the Author):

Authors have adequately addressed the questions raised in the first revision as well as performed new experimental mouse models.

Response: Thanks for all your efforts throughout this review process.

Minor Comments

- Western Blottings were not quantified in Figure 4E. In order to properly conclude immunoblottings need to be quantified.

Response: Thank you for your suggestion. As the reviewer suggested, we further quantified the results of western blots in Figure 4E (Supporting Fig. 1).

Supporting Fig. 1 The quantification of western blots in Figure 4E.

- Authors have incorporated a new figure as Figure 3E. In this Figure the total tumor number and the number of tumors with more than 3mm diameter in the liver of DEN+CCl4 mice treated with anti-CCL2 were included. Please in order to be consistent with all the figures please include above the two graphs in panel E that this mice are DEN+CCl4. Same applies for Figure 2E. in line with this, in those panels in which tumor volume and tumor weight are included, include the mouse model above both graphs or in one of them but it needs to be consistent in all the figures. Some examples include Figure 2F, Figure 2G, Figure 2H, Figure 3F and Figure 3H.

Response: Thank you for your suggestion. Accordingly, we have further labeled the titles of Figure 2F, Figure 2G, Figure 2H, Figure 3E, Figure 3F and Figure 3H to avoid any misleading.

Reviewer #3 (Remarks to the Author):

Overall the authors did a significant attempt to address my comments. I am still puzzled by the lack of effects on NK but the new data is supportive of this statement. It will be helpful to expand the discussion of this point.

Response: Thanks for all your efforts throughout this review process.

For supp Fig. 15 it will be helpful to get an insert of a magnified image.

Response: Thank you for your suggestion. However, due to the limitation of the instrument, we have already provided the largest magnified image (Original magnification 20 x 10, Scale bar, 100 μ m).